# Antagonist actions of CMK-1/CaMKI and TAX-6/calcineurin along the *C. elegans* thermal avoidance circuit orchestrate adaptation of nociceptive response to repeated stimuli

**Martina Rudgalvyte[1], Zehan Hu[1], Dieter Kressler[1,2], Jörn Dengjel[1], Dominique A Glauser[1]***

[1]Department of Biology, University of Fribourg, Fribourg, Switzerland; [2]Metabolomics and Proteomics Platform (MAPP), Department of Biology, University of Fribourg, Fribourg, Switzerland

**\*For correspondence:**
dominique.glauser@unifr.ch

**Competing interest:** The authors declare that no competing interests exist.

## eLife Assessment

This study uses *C. elegans* to investigate how the Calcium/Calmodulin-dependent kinase CMK-1 regulates adaptation to thermo-nociceptive stimuli. The authors use **compelling** approaches to identify Calcineurin as a phosphorylation target of CMK-1 and to investigate the relationship between CMK-1 and Calcineurin using gain and loss of function genetic and pharmacological methods. The findings of this study are **valuable** as they show that CMK-1 and Calcineurin act in separate neurons in an antagonistic and complex manner to regulate thermo-nociceptive adaptation, and these results may be relevant for understanding some chronic human pain conditions.

**Abstract** Thermal nociception in *Caenorhabditis elegans* is regulated by the $Ca^{2+}$/calmodulin-dependent protein kinase CMK-1, but its downstream effectors have remained unclear. Here, we combined in vitro kinase assays with mass-spectrometry-based phosphoproteomics to identify hundreds of CMK-1 substrates, including the calcineurin A subunit TAX-6, phosphorylated within its conserved regulatory domain. Genetic and pharmacological analyses reveal multiple antagonistic interactions between CMK-1 and calcineurin signaling in modulating both naive thermal responsiveness and adaptation to repeated noxious stimuli. Cell-specific manipulations indicate that CMK-1 acts in AFD and ASER thermo-sensory neurons, while TAX-6 functions in FLP thermo-sensory neurons and downstream interneurons. Since CMK-1 and TAX-6 act in distinct cell types, the phosphorylation observed in vitro might not directly underlie the behavioral phenotype. Instead, the opposing effects seem to arise from their distributed roles within the sensory circuit. Overall, our study provides (1) a resource of candidate CMK-1 targets for further dissecting CaM kinase signaling and (2) evidence of a previously unrecognized, circuit-level antagonism between CMK-1 and calcineurin pathways. These findings highlight a complex interplay of signaling modules that modulate thermal nociception and adaptation, offering new insights into potentially conserved mechanisms that shape nociceptive plasticity and pain (de)sensitization in more complex nervous systems.

## Introduction

Animals from invertebrates to mammals can detect and avoid harmful stimuli, which is essential for their survival and well-being. Noxious stimuli are detected and encoded by specialized primary sensory neurons in a process called nociception (*Dubin and Patapoutian, 2010*). Nociceptive signals are relayed in the nervous system via synaptic connections to be further processed and trigger different responses, such as behavioral changes and the perception of pain (*Treede, 1995*). Pain sensitivity is not fixed and can be adjusted in various physiological and pathological contexts through nociceptive plasticity processes such as hyperalgesia or nociceptive desensitization/habituation (*Kidd and Urban, 2001*; *Sandkühler, 2009*; *Rodriguez-Raecke et al., 2010*; *Breimhorst et al., 2012*; *May et al., 2012*; *O'Neill et al., 2012*, *Basith et al., 2016*). Prolonged or repeated exposure to noxious stimuli can lead to either sensitization or desensitization/habituation depending on the stimuli intensity, frequency, and various physiological factors (*Treede, 1995*). Because impaired nociceptive habituation is linked to various human chronic pain conditions (*de Tommaso et al., 2005*; *Smith et al., 2008*; *de Tommaso et al., 2011*), obtaining a deeper understanding of the cellular and molecular processes at play appears highly relevant to aid in the development of new therapeutic strategies for pain management. Several molecular mechanisms have been shown to modulate nociceptive pathways from the periphery to the brain (*Kuner and Flor, 2017*; *Pace et al., 2018*; *Khan et al., 2019*), which involve the regulated activity of proteins such as membrane receptors or ion channels (involved in sensory transduction, cell excitability, or signal conduction). Many kinases and phosphatases, which actuate protein regulation via the control of their phosphorylation status and that were previously shown to broadly regulate neural plasticity in the nervous system, are also involved in nociceptive plasticity (*Willis, 2001*; *Fu et al., 2008*; *Wang and Zhang, 2012*; *Isensee et al., 2014*; *Pace et al., 2018*). These regulatory pathways include intracellular signaling by calcium/calmodulin-dependent protein kinases (CaMKs) (*Shum et al., 2005*; *Liang et al., 2012*; *Schild et al., 2014*; *Zhou et al., 2017*), which might couple cellular activity levels with pleiotropic intracellular effects via post-translational modifications over a large repertoire of potential target proteins. Identifying the phosphorylation substrates of these kinases that act in the nociceptive pathway to mediate plasticity effects could provide relevant insight for future pain management translational research.

The nematode *Caenorhabditis elegans* has emerged as a powerful model to study the molecular and cellular bases of nociception and its plasticity. *C. elegans* produces innate avoidance behaviors in response to a variety of noxious stimuli, including irritant chemicals, harsh touch and noxious heat (*Kaplan and Horvitz, 1993*; *Wittenburg and Baumeister, 1999*; *Hilliard et al., 2005*; *Li et al., 2011*; *Liu et al., 2012*). Noxious heat stimuli targeting the animal head or the entire animal trigger a stereotyped reversal behavior (*Wittenburg and Baumeister, 1999*; *Liu et al., 2012*; *Byrne Rodgers and Ryu, 2020*; *Lia and Glauser, 2020*), which involves several thermo-sensory neurons, including AFD, AWC, and FLP, proposed to work as thermo-nociceptors (*Chatzigeorgiou and Schafer, 2011*; *Liu et al., 2012*; *Kotera et al., 2016*). The molecular components underpinning the function of the nociceptive system are well-conserved. For example, transient receptor potential (TRP) channels mediate worm thermo-nociceptive responses, like in fly and mammals (*Chatzigeorgiou et al., 2010*; *Glauser et al., 2011*; *Liu et al., 2012*; *Nkambeu et al., 2020*). Furthermore, persistent or repeated noxious heat stimuli cause a progressive reduction in the rate of heat-evoked reversals (*Lia and Glauser, 2020*; *Jordan and Glauser, 2023*), which we will refer to here as thermo-nociceptive adaptation. A genetic screen for human pain-associated gene orthologs revealed thermo-nociceptive adaptation alterations for many corresponding worm mutants, substantiating the interest of the model (*Jordan and Glauser, 2023*). Among conserved molecular players, the worm CaMKI (named CMK-1) was shown to mediate thermo-nociceptive adaptation (*Schild et al., 2014*; *Lia and Glauser, 2020*; *Jordan and Glauser, 2023*). CMK-1 is broadly expressed in the worm nervous system and, beyond thermo-nociceptive adaptation, controls multiple experience-dependent plasticity processes, such as those related to developmental trajectories (*Neal et al., 2015*), the encoding of preferred growth temperature (*Yu et al., 2014*), salt aversive learning (*Lim et al., 2018*), and habituation to repeated touch stimuli (*Ardiel et al., 2018*). CMK-1 was shown to regulate the expression of the *guanylyl cyclase-8* (*gcy-8*) gene (*Satterlee et al., 2004*), the AMPA glutamate receptor-1 (*glr-1*) gene (*Moss et al., 2016*), the ortholog of the human DACH1/2 Dachsund transcription factor gene (*dac-1*) and the PY domain transmembrane protein-1 (*pyt-1*) gene (*Harris et al., 2023*). The impact of CMK-1 on gene transcription is partially mediated by

the CREB homolog-1 (CRH-1) transcription factor (*Harris et al., 2023*), which could be phosphorylated by CMK-1 in vitro (*Kimura et al., 2002*) and in a heterologous expression system (*Eto et al., 1999*). A previous study used an in silico approach to systematically predict potential CMK-1 phosphorylation substrates and revealed repeated touch habituation phenotypes in mutants for several candidate targets (*Ardiel et al., 2018*). Apart from these few studies, we still know very little on the CMK-1 downstream effectors that mediate the numerous biological actions of CMK-1, including in mediating thermo-nociceptive adaptation.

Calcineurin is a well-conserved eukaryotic $Ca^{2+}$/calmodulin activated serine/threonine phosphatase (*Klee et al., 1979*; *Rusnak and Mertz, 2000*), involved in $Ca^{2+}$ signaling pathway and functioning in several tissues, such as muscles, T cells, and neurons, where it regulates synaptic plasticity and memory (*Groth et al., 2003*; *Mukherjee and Soto, 2011*). Calcineurin regulates the activity of numerous proteins, such as transcription factors (*Molkentin, 2004*), receptors, mitochondrial proteins, and microtubules depending on $Ca^{2+}$ signaling state (*Peuker et al., 2022*). Overexpression of calcineurin can cause cardiac hypertrophy (*Gillis et al., 2004*; *Yuan et al., 2024*), higher infarct volumes (*Dai et al., 2024*), whereas lack of function can cause defects in kidney development (*Gooch et al., 2004*), and an increase in autophagy (*Ke et al., 2023*). Chronic inhibition of calcineurin with immunosuppressant drugs, like Cyclosporin A, was shown to cause irreversible neuropathic pain in some patients, a phenomenon referred to as calcineurin inhibitor-induced pain syndrome (*Smith, 2009*). Several studies in mice have highlighted that calcineurin signaling modulate thermal nociception (*Sato et al., 2007*). Several pathways working downstream of calcineurin signaling have been proposed to mediate the regulation of nociception, including NFAT transcription factors, TWIK-related spinal cord potassium channels (TRESK), TRP channels, and $\alpha 2\delta$-1 $Ca^{2+}$ channels (*Smith, 2009*; *Huang et al., 2020*; *Huang et al., 2022*). Nevertheless, we still have a cursory understanding of how calcineurin signaling integrates with other intracellular signaling pathways at different loci in the nociceptive circuit to orchestrate nociceptive plasticity.

Calcineurin is a heterodimeric protein consisting of one calcineurin A (CnA) catalytic subunit and one calcineurin B (CnB) regulatory subunit. CnA contains catalytic and autoinhibitory domains, as well as CnB and $Ca^{2+}$/calmodulin-binding domains. In the absence of $Ca^{2+}$ signaling, the autoinhibitory domain suppresses catalytic phosphatase activity. Increase in cytosolic $Ca^{2+}$ promotes $Ca^{2+}$ binding to CnB and to CaM. $Ca^{2+}$/CnB and $Ca^{2+}$/CaM in turn bind to CnA to release the autoinhibition, allowing substrates to access the active site and thereby the activation of CnA (*Bandyopadhyay et al., 2002*; *Kuhara et al., 2002*; *Wang et al., 2008*). The *C. elegans* genome encodes one CnA homolog, *tax-6* (aka *cna-1*), and one CnB homolog, *cnb-1*, displaying conserved structural and biochemical features with vertebrate calcineurin proteins (*Bandyopadhyay et al., 2002*; *Bandyopadhyay et al., 2004*). Lack of *tax-6* function causes thermal hypersensitivity and increased osmosensation, as well as increased adaptation of the olfactory system (*Kuhara et al., 2002*). Loss-of-function mutants display a thermophilic behavior, and this anomaly could be rescued by wild-type *tax-6* expression in AFD neurons. *tax-6* gain-of-function mutants show defective enteric muscle contraction (*Lee et al., 2005*), as well as higher brood size and hypersensitivity to serotonin (*Lee et al., 2004*). The loss of *cnb-1* causes lethargic movements, delayed egg laying, and reduced growth (*Bandyopadhyay et al., 2002*).

Here, we combined in vitro kinase assays with shotgun phosphoproteomics to empirically identify CMK-1 phospho-substrates in worm peptide and protein libraries. CMK-1 was found to phosphorylate TAX-6/CnA in a highly conserved regulatory region, suggesting cross-regulation between CaMK and calcineurin signaling in worms. We next combined genetic and pharmacological manipulations with a quantification of noxious heat reversals to assess the role of CMK-1 and TAX-6 signaling and their interactions in controlling naive animal responsiveness and the adaptation to repeated stimulation. These follow-up analyses confirmed a complex set of antagonistic signaling actions produced by the two pathways. However, the two intracellular signaling pathways appear to primarily function in separate neuron types to control thermo-nociceptive adaptation, indicating that the phosphorylation of TAX-6/CnA by CMK-1 might not be relevant in vivo for the control of thermo-nociception. Collectively, our results reveal multiple direct and indirect interactions between CaMK and calcineurin signaling and pave the way for deeper studies on the molecular control of nociceptive adaptation in a simple genetic model.

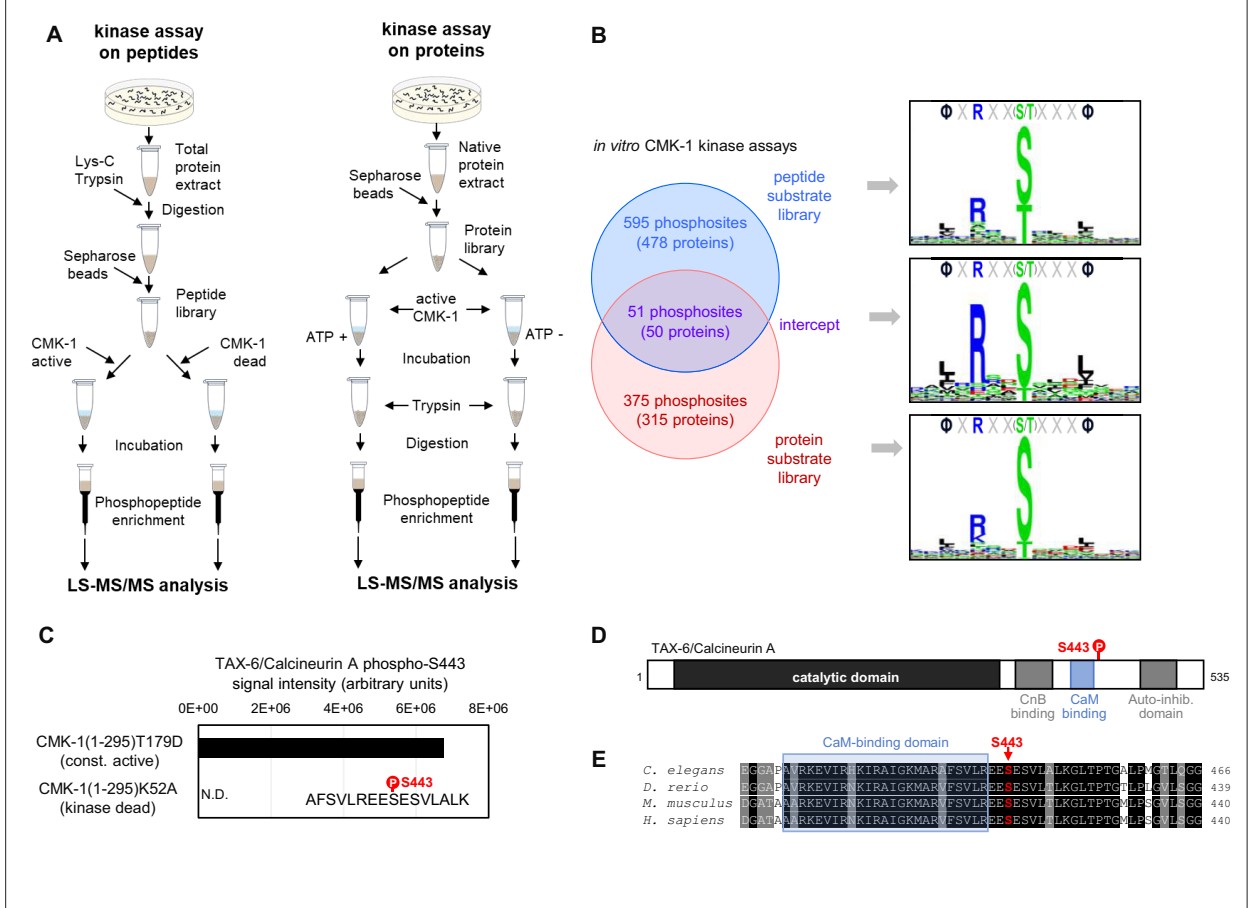

**Figure 1.** Identification of CMK-1 phosphorylation substrates. (**A**) Schematic of the two approaches used for the large-scale identification of the CMK-1 phosphorylation substrates in vitro. In vitro kinase assays using a *C. elegans* peptide library (left) or a *C. elegans* native protein library (right) were followed by MS-based phosphoproteomics analyses. The activity of a constitutively active mutant (CMK-1(1–295)T179D) was compared to control situations with either kinase dead mutant (CMK-1(1–295)K52A) or without ATP co-substrate during the incubation. (**B**) Comparison of the results with the two approaches, showing the number of overlapping and non-overlapping phosphosites, the number of corresponding proteins, as well as the 15-residue consensus sequence surrounding the phosphosites. Logos created with Seq2Logo. (**C**) Results of a separate in vitro kinase assay in which purified TAX-6/CnA was used as substrate and showing TAX-6/CnA S443 phosphorylation. N.D., not detected. (**D**) Diagram presenting the different regions of the *C. elegans* TAX6/CnA protein and the localization of S443. (**E**) Alignments showing high conservation of a CnA protein region around S443 across *C. elegans*, *Danio rerio*, *Mus musculus*, and *Homo sapiens* (Uniprot assessions: Q0G819-2|PP2B_CAEEL; A3KGZ6|A3KGZ6_DANRE; P63328|PP2BA_MOUSE; Q08209|PP2BA_HUMAN).

The online version of this article includes the following figure supplement(s) for figure 1:

**Figure supplement 1.** Comparison of empirically identified CMK-1 phospho-substrates with previously published predictions.

## Results

### CMK-1 phosphorylates multiple substrates in vitro, including TAX-6/CnA

To identify phosphorylation substrates of CMK-1, we performed in vitro kinase assays on two types of substrate libraries followed by mass-spectrometry-based phosphoproteomics (*Figure 1A*). To that end, we purified recombinant CMK-1(1–295)T179D mutant protein produced in *E. coli*. This mutant is a constitutively active form of CMK-1 that lacks the C-terminal auto-inhibitory domain and harbors a T179D phosphomimic mutation, thus bypassing the need for $Ca^{2+}$/CaM binding and for phosphorylation by CKK-1. In two separate experiments, we assessed the phosphorylation produced by CMK-1(1–295)T179D, first, on a library of worm peptides produced by trypsin digestion of whole protein extracts and, second, on a whole protein library produced in non-denaturing conditions. As expected given the strong structural and functional conservation within the CaM kinase family, the consensus

substrate motifs obtained with each substrate library matched the ΦXRXX(S/T)XXXΦ consensus previously characterized for mammalian CaMK (where Φ represents hydrophobic residues, *Figure 1B*; *Lee et al., 1994*; *White et al., 1998*). This result supports the validity of our in vitro kinase assays to identify direct CMK-1 targets and indicates that our experimental design efficiently mitigated the impact of any potential kinase activity coming from *E. coli* protein contamination or from worm kinases in the case of whole protein extracts.

With the peptide library data, we identified 646 phosphosites in 478 proteins (*Supplementary file 1*). With the whole protein library, we identified 427 phosphosites in 365 proteins (*Supplementary file 1*). Fifty-one phosphosites in 50 proteins were common between the two datasets (*Supplementary file 1*, *Figure 1B*). As more extensively discussed later in the text (see Discussion section), these datasets are expected to include both phosphosites that are relevant CMK-1 targets in vivo and 'false positive' phosphosites which are not relevant targets in vivo. These datasets are also expected to be biased toward abundant proteins, which are more likely to be detected. Consistent with this expectation, the most significant enrichment found via gene ontology (GO) term analyses is related to ribosomal and cytoskeletal proteins (*Supplementary file 2*). Outside of these categories and among the phosphosites common to our two datasets, we were intrigued by the presence of a CMK-1 target phosphosite on serine 443 of TAX-6/CnA. This phosphosite is located in a highly conserved region of TAX-6/CnA regulatory domain, just on the C-terminal side of the CaM-binding domain (*Figure 1D*). To confirm that CMK-1 can phosphorylate TAX-6/CnA S443 in vitro, we repeated the kinase assay using recombinant TAX-6/CnA protein purified from *E. coli* as substrate. Abundant phosphorylation of S443 was observed upon treatment with the constitutively active CMK-1 form, whereas this phosphorylation remained undetected with kinase-dead control treatment (*Figure 1C*). We conclude that CMK-1 can phosphorylate TAX-6/CnA on S443 in vitro.

## Thermo-nociceptive adaptation is impaired by CMK-1 down-regulation and by both up- and down-regulation of TAX-6/CnA

We previously reported that CMK-1 controls the adaptation to persistent noxious heat stimulations (*Schild et al., 2014*) or repeated stimulations (*Lia and Glauser, 2020*). This adaptation effect consists in a progressive reduction of the noxious heat-evoked reversal response observed in wild type during 1 hr of repeated heat stimulation. Importantly, this adaptation effect does not result from thermal damages or from an exhaustion of neuronal or muscular tissues, as evidenced by the existence of non-adapting mutants which can maintain a constant response level (*Jordan and Glauser, 2023*). We confirmed these previous observations by quantifying heat-evoked reversals in naive animals (T0) and animals submitted to a series of repeated heat stimulations for 1 hr (T60, *adaptation treatment* with an interstimulus interval (ISI) of 20 s, *Figure 2A*). Whereas wild-type response is significantly decreased following repeated stimulation treatment (1 hr adaptation, T60), this adaptation effect is absent in a *cmk-1(ok287)* loss-of-function mutants (*Figure 2B*). Conversely, a *cmk-1(syb1633)* gain-of-function mutant with a CMK-1(T179D) over-activating mutation promoted a faster adaptation (*Figure 2B*, *Figure 2—figure supplement 1*). Therefore, our data confirm that CMK-1 activity is necessary and sufficient to promote thermo-nociceptive adaptation.

Since TAX-6/CnA is a CMK-1 kinase substrate in vitro, we hypothesized that it could take part in the regulation of thermo-nociceptive adaptation and evaluated the impact of genetic and pharmacological manipulations down- or up-regulating calcineurin signaling activity. The permanent loss of TAX-6/CnA in *tax-6(p675)* mutants or of CnB/CNB-1 in a *cnb-1(jh103)* and *cnb-1(ok276)* mutants caused a marked elevation in the rate of spontaneous reversals (*Figure 2—figure supplement 2*). While this observation indicates that TAX-6/CnA signaling regulates reversal behavior, the elevation was so high that it precluded quantifying heat-evoked reversals and plasticity response. We next turned to a pharmacological approach using the calcineurin inhibitor Cyclosporin A (*Liu et al., 1991*). Treating wild-type worms with 10 mM Cyclosporin A for 24 hr prior to behavioral assays did not cause the problematic elevated spontaneous reversal phenotype, but significantly reduced the thermal adaptation effect (*Figure 2C*). An intact calcineurin signaling is therefore required for thermo-nociceptive adaptation. Next, we examined the impact of over-activating TAX-6/CnA, using a *tax-6(jh107)* gain-of-function mutant lacking the auto-inhibitory C-terminal domain of TAX-6/CnA, which we will refer to as *tax-6(gf)*. We noted a very slightly enhanced reversal response in naive *tax-6(gf)* mutants in comparison to wild type (*Figure 2D*). More strikingly, the adaptation effect in *tax-6(gf)* mutants was strongly

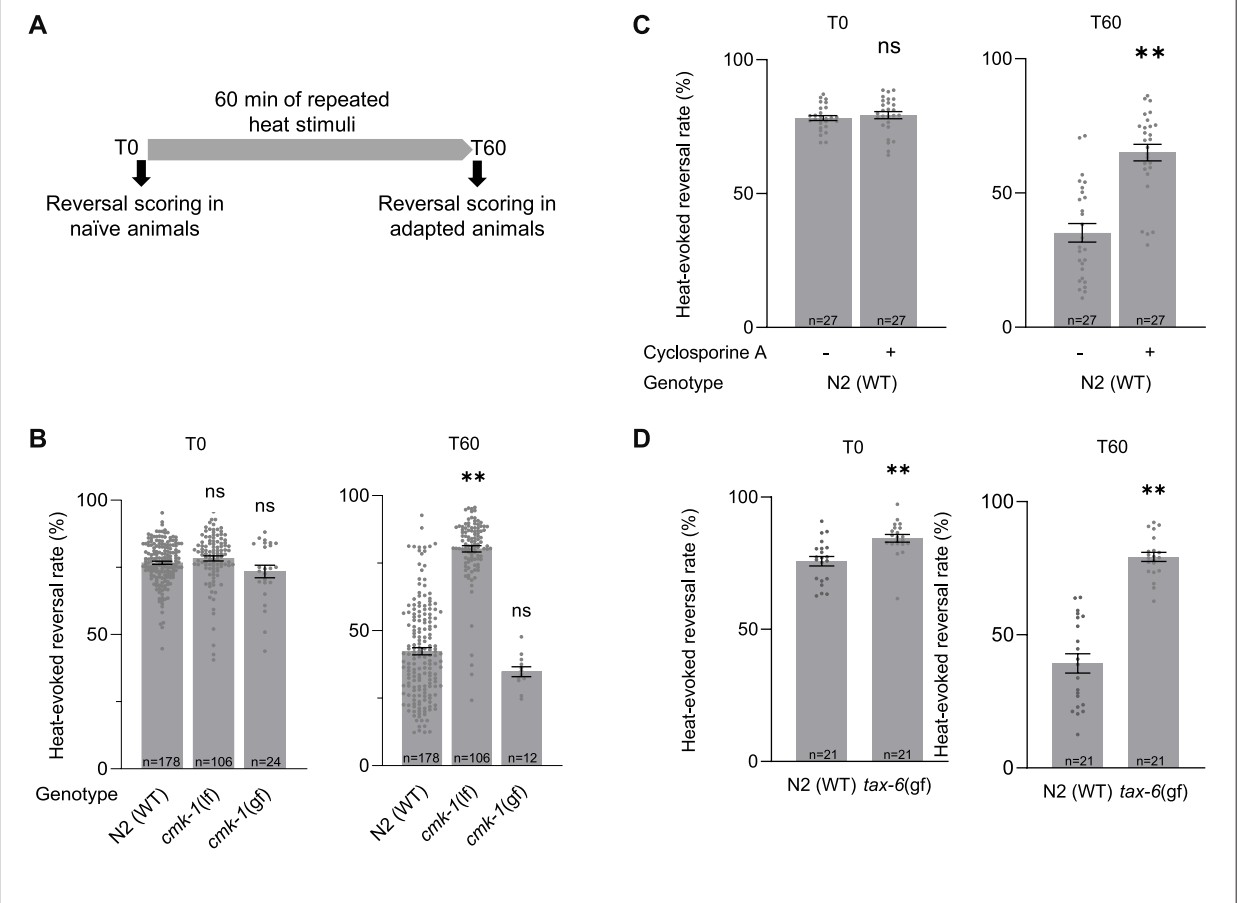

**Figure 2.** Impact of loss and gain of CMK-1 and TAX-6/CnA function on *C. elegans* thermo-nociceptive response. (**A**) Schematic of the scoring procedure. Heat-evoked reversals were first scored in naive adult *C. elegans* that had never been exposed to thermal stimuli (T0), animals exposed to 4 s heat pulses every 20 s during 60 min, prior to an endpoint scoring after adaptation (T60). (**B–D**) Heat-evoked reversal scored in the indicated genotypes. Results as fraction of reversing animals. Each point corresponds to one assay scoring at least 50 animals. Average (gray bars) and SEM (error bars) with indicated *n* representing the number of independent assays. *cmk-1(gf)* is *cmk-1(syb1633)*. *cmk-1(lf)* is *cmk-1(ok287)*. *tax-6(gf)* is *tax-(j107)*. For calcineurin inhibition, 10 µM Cyclosporin A was used 24 prior to experiments. **p < 0.01 versus N2(WT) control in the specific condition by Bonferroni–Holm post hoc tests. ns, not significant.

The online version of this article includes the following source data and figure supplement(s) for figure 2:

**Source data 1.** Numerical data and p values presented in the figures.

**Figure supplement 1.** CMK-1 overactivating mutation T179D accelerates thermo-nociceptive adaptation.

**Figure supplement 1—source data 1.** Numerical data and p values presented in the figures.

**Figure supplement 2.** Spontaneous reversal rate in wild type and different single and double mutants.

impaired. Therefore, our data show that both overactivation and inhibition of TAX-6/CnA signaling can block thermo-nociceptive adaptation.

In summary, gain- and loss-of-function analyses highlight a CMK-1-dependent pathway promoting and potentially two antagonistic TAX-6/CnA-dependent pathways that promote and inhibit thermo-nociceptive adaptation, respectively.

## CMK-1 and TAX-6/CnA signaling regulate thermo-nociceptive adaptation through a set of inhibitory cross-talks

To assess the potential interactions between CMK-1 and calcineurin signaling in the control of thermo-nociceptive adaptation, we systematically tested combinations of up- or down-regulating manipulations in the two pathways.

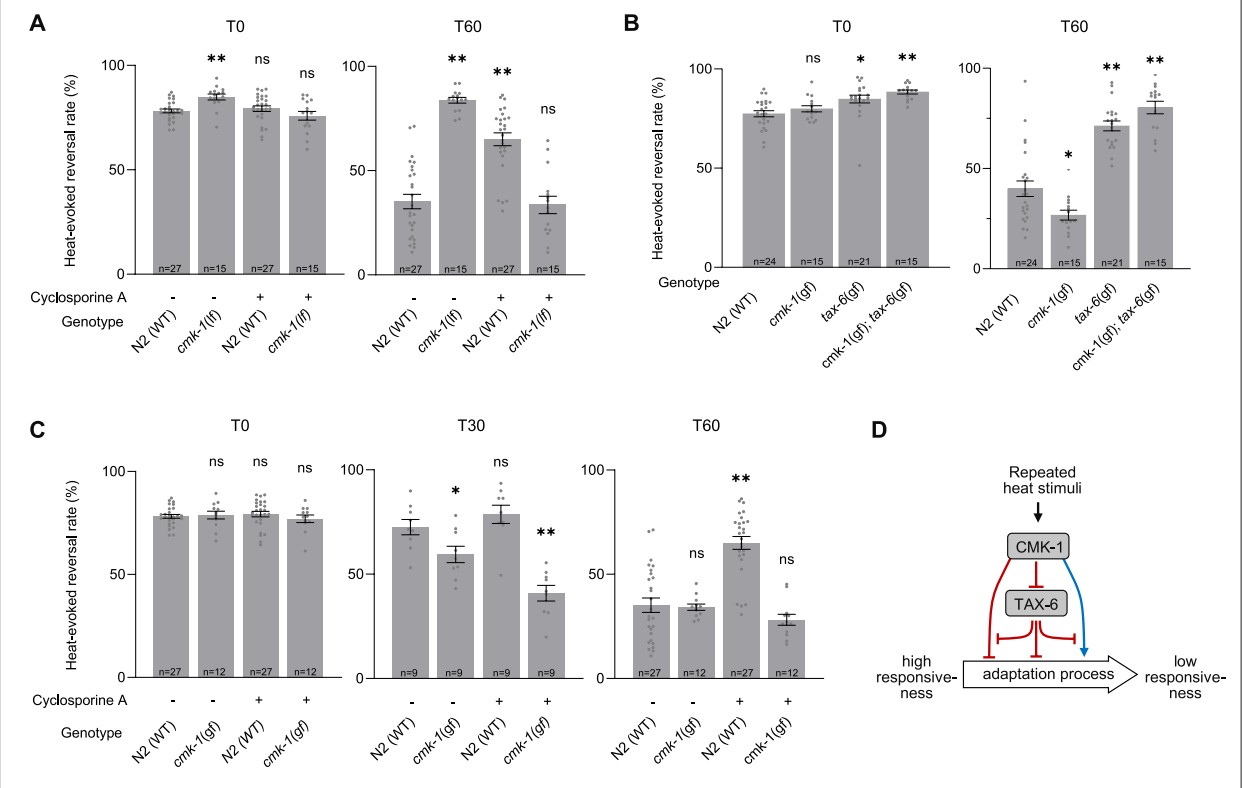

**Figure 3.** Functional interactions between CMK-1 and TAX-6/CnA in the regulation of thermo-nociceptive adaptation. (**A–C**) Assessment of the impact of joint gain- and loss-of-function manipulations affecting the CMK-1 and TAX-6/CnA pathways. Heat-evoked response in naive animals (T0) and after 60 min of repeated stimulations (T60), scored and reported as in **Figure 2**. **\*\*p < 0.01 and \*p < 0.05 versus N2(WT) control in the specific condition by Bonferroni–Holm post hoc tests. ns, not significant. (**D**) Schematic of a model explaining the multiple antagonistic interactions observed between CMK-1 and TAX-6/CnA signaling. cmk-1(gf) is cmk-1(syb1633) harboring the CMK-1(T179D) overactivating mutation. cmk-1(lf) is cmk-1(ok287). tax-6(gf) is tax-6(jh107).

The online version of this article includes the following source data and figure supplement(s) for figure 3:

**Source data 1.** Numerical data and p values presented in the figures.

**Figure supplement 1.** Model of the multiple antagonistic interactions observed between CMK-1 and TAX-6/CnA signaling.

First, we examined the impact of down-regulating both CMK-1 and TAX-6/CnA signaling by treating cmk-1(lf) mutants with Cyclosporin A. Surprisingly, whereas separate down-regulation of either CMK-1 or TAX-6/CnA pathway strongly blocked adaptation, their joint down-regulation caused animals to adapt like wild type (**Figure 3A**). An intact adaptation when both pathways are inhibited supports the notion that CMK-1 and TAX-6/CnA represent regulators of one or more adaptation pathways that can also operate independently. In addition, this observation supports a model in which (1) the anti-adaptation effect caused by CMK-1 inhibition is mediated by TAX-6/CnA and, conversely, (2) that the anti-adaptation effect caused by TAX-6/CnA inhibition requires intact CMK-1 activity.

Second, we tested the impact of simultaneously activating CMK-1 and calcineurin pathways by examining the behavior of cmk-1(gf);tax-6(gf) double mutants. We found that the double mutants behaved like tax-6(gf) single, entirely lacking adaptation (**Figure 3B**). Therefore, the pro-adaptation effect of the CMK-1(T179D) activating mutation is fully blocked by TAX-6/CnA overactivation, which is compatible with a model in which CMK-1 over-activation might work by inhibiting calcineurin signaling.

Third, we assessed the impact of concomitantly up-regulating calcineurin and down-regulating CMK-1 signaling, by examining the behavior of cmk-1(lf);tax-6(gf) double mutants. Whereas the main phenotypic feature in each single mutant was a lack of adaptation, the mutation combination produced a synthetic effect massively elevating spontaneous reversal, which precluded quantifying heat-evoked reversals (**Figure 2—figure supplement 2**).

Fourth, we tested the effect of concomitantly up-regulating CMK-1 activity and inhibiting calcineurin signaling by treating *cmk-1(gf)* mutants with Cyclosporin A. We found that, unlike in the *cmk-1(wt)* background, Cyclosporin A treatment in *cmk-1(gf)* mutants did not prevent thermo-nociceptive adaptation (*Figure 3C*). The fact that an overactive CMK-1 signaling can compensate for the inhibition of TAX-6/CnA suggests that the anti-adaptation effect of calcineurin inhibition involves the ability to down-regulate CMK-1 signaling. Furthermore, the ability of CMK-1 overactivation to promote adaptation when calcineurin is inhibited indicates that at least one CMK-1 regulatory branch works independently or downstream of calcineurin signaling.

Collectively, our results do not support a simple linear model in which CMK-1 would only work upstream of TAX-6/CnA. Instead, our data suggest a model in which thermo-nociceptive adaptation processes are regulated via the antagonist actions of CMK-1 and TAX-6/CnA signaling operating through a non-linear inhibitory network, such as the one depicted in *Figure 3D* and discussed below (see Discussion section).

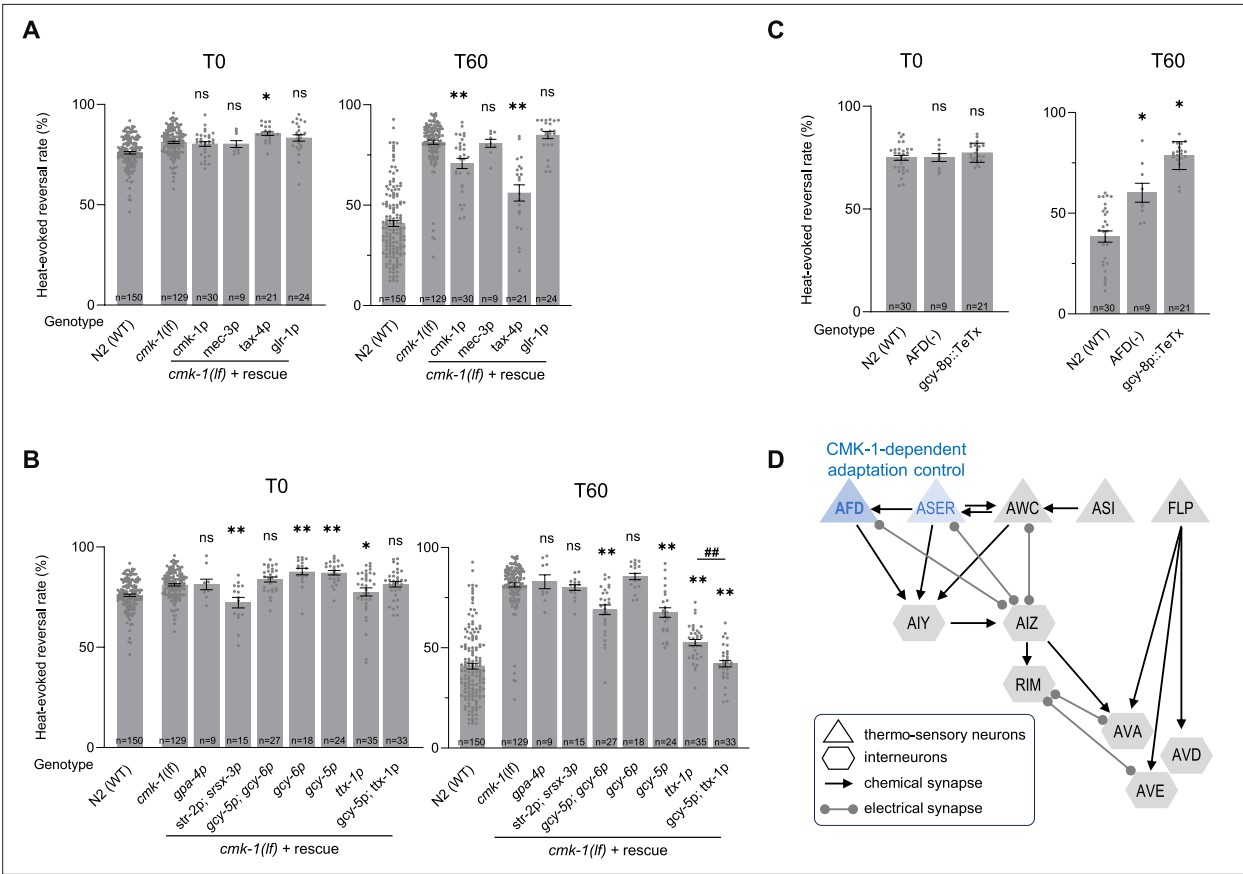

**Figure 4.** CMK-1 works in AFD and ASER to control thermo-nociceptive adaptation. (**A, B**) Determination of the CMK-1 place of action in the control of thermo-nociceptive adaptation, using cell-specific rescue in *cmk-1(ok287)* (*cmk-1(lf)*) background. Heat-evoked response in naive animals (T0) and after 60 min of repeated stimulations (T60), scored and reported as in *Figure 2*. The promoters used to restore CMK-1 expression are indicated below each bar. **p < 0.01 and *p < 0.05 versus *cmk-1(lf)* by Bonferroni–Holm post hoc tests. ns, not significant. ##p < 0.01 for the specific contrast between *ttx-1p* and the *gcy-5p;ttx-1p* combination. N2(WT) data are shown for comparison purpose. (**C**) Impact of genetic manipulation ablating AFD with a caspase construct (AFD(−)) or inhibiting AFD neurotransmission with TeTx heterologous expression. **p < 0.01 and *p < 0.05 versus N2(WT) control by Bonferroni–Holm post hoc tests. (**D**) Schematic of the hypothetical circuit controlling noxious-heat-evoked reversals, including the *tax-4*-expressing thermo-responsive sensory neurons AFD, AWC, ASER, and ASI, the *mec-3*-expressing FLP thermo-nociceptor, and a subset of downstream interneurons known to mediate reversal response, including the *glr-1*-expressing RIM, AVA, AVD, and AVE interneurons. The main loci of action of CMK-1 determined through the cell-specific rescue approach are highlighted in blue.

The online version of this article includes the following source data for figure 4:

**Source data 1.** Numerical data and p values presented in the figures.

## CMK-1 acts in AFD and ASER sensory neurons to promote thermo-nociceptive adaptation

Our next goal was to assess the place of action of CMK-1 signaling in the promotion of thermo-nociceptive adaptation. To that end, we used a neuron-specific rescue approach in the *cmk-1(lf)* background. We started our experiments by restoring CMK-1 expression in candidate thermo-sensory neurons, using *mec-3p* promoter to drive expression in FLP and *tax-4p* promoter to drive expression in AFD, AWC, ASI, and ASE, as well as *glr-1p* promoter to drive expression in a subset of interneurons known to mediate reversal response. A positive control using *cmk-1p* promoter showed a partial, yet significant rescue effect (*Figure 4A*). We observed a strong rescue effect with the *tax-4p* promoter, but no rescue effect with *mec-3p* or *glr-1p* (*Figure 4A*). While not ruling out a contribution of other neurons, these data point to *tax-4*-expressing neurons as a major place of action for CMK-1 in the control of thermo-nociceptive adaptation.

To further dissect the contribution of *tax-4*-expressing neurons, we conducted additional rescue experiments using *gpa-4p* to target ASI, a combination of *str-2p* and *srsx-3p* to target the two AWC neurons (AWCon and AWCoff, respectively), a combination of *gcy-5p* and *gcy-6p* to target ASEL and ASER neurons, *gcy-6* alone to target ASEL, *gcy-5p* alone to target ASER and *ttx-1p* to target AFD. We observed an almost total rescue effect with the *ttx-1p* promoter (AFD) and a more partial rescue effect with either *gcy-5p+gcy-6p* (both ASEL and ASER) or *gcy-5p* alone (only ASER) (*Figure 4B*). No rescue effect was observed with the other promoters. We additionally tested the combined use of *ttx-1p* and *gcy-5p* to target both AFD and ASER and obtained a maximal rescue effect even stronger than with *ttx-1p* alone (*Figure 4B*). Taken together, our data point to AFD as a major (and to ASER as a more minor) place of action for CMK-1 signaling in the control of thermo-nociceptive adaptation to repeated heat-stimuli.

To further assess the role of AFD neurons, we evaluated the behavior of animals with genetically ablated AFD neurons. Interestingly, AFD neurons were dispensable for heat-evoked reversal responses in naive animals under our experimental conditions, but required for the thermo-nociceptive adaptation (*Figure 4C*). We observed similar results when selectively inhibiting AFD neurotransmission in animals expressing the tetanus toxin (TeTx) in AFD (*Figure 4C*).

Taken together, our results show that AFD neurons mediate thermo-nociceptive adaptation and that intact CMK-1 signaling and neurotransmission in these neurons is essential for this process (*Figure 4D*). ASER might represent an additional locus of CMK-1 action with a milder quantitative contribution to the process (*Figure 4D*).

## TAX-6/CnA activity in RIM and command interneurons inhibits thermo-nociceptive adaptation

In order to assess the cellular place of action of TAX-6/CnA in the control of thermo-nociceptive adaptation, we used a neuron-type-specific overactivation approach. We created transgenes containing a *tax-6(gf)* cDNA to drive the expression of the overactive, truncated form of TAX-6/CnA encoded in the *tax-6(jh107)* mutant. Like for CMK-1 rescue experiments above, we started by targeting candidate thermo-sensory neurons using either the *mec-3p* (FLP) or *tax-4p* (AFD, AWC, ASI, and ASE) promoters, as well as interneurons with the *glr-1p* promoter. We examined the ability of the transgenes to replicate the two noticeable phenotypes of *tax-6(gf)* mutants: a slightly enhanced response in naive animals and a strong impairment of adaptation.

First, regarding naive animal response, we found that *[mec-3p::tax-6(gf)]* and, to a lesser extent *[glr-1p::tax-6(gf)]*, transgenes caused a slight elevation in the heat-evoked response, whereas *[tax-4p:tax-6(gf)]* did not produce any effect (*Figure 5A*). *glr-1p* promoter drives expression in several interneurons, including AVA, AVD, AVE, and RIM whose activity is required and sufficient to trigger reversals (*Alkema et al., 2005*; *Gray et al., 2005*; *Guo et al., 2009*). To further dissect the specific interneurons involved, we expressed the *tax-6(gf)* transgene in AVA, AVD, and AVE using the *nmr-1p* promoter, or the *lgc-39p* promoter, in RIM (and RIC) using the *cex-1p* promoter, and only in RIM using the *tdc-1p* promoter (*Brockie et al., 2001*; *Lemieux et al., 2015*; *Thapliyal et al., 2023*). With all four promoters, we observed an effect similar to that with *glr-1p*. Taken together, these results indicate that over-activating calcineurin signaling in *mec-3p*-expressing neurons (most likely in FLP thermo-nociceptor) or in a subset of reversal-mediating interneurons such as RIM or AVA/AVD/AVE is sufficient to up-regulate noxious heat-responsiveness in naive animals (*Figure 5C*, top).

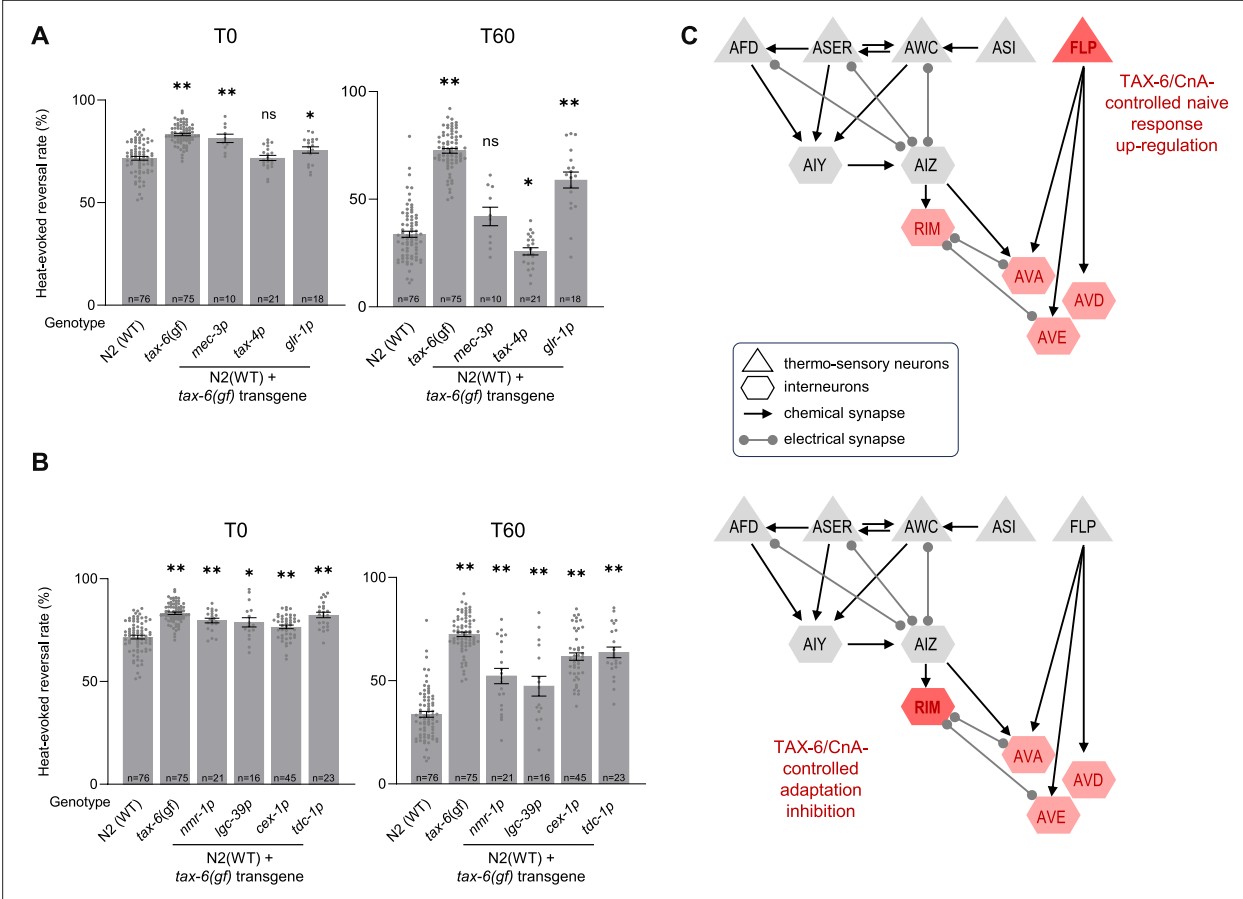

**Figure 5.** TAX-6/CnA activity in RIM and AVA/AVD/AVE inhibit thermo-nociceptive adaptation. (**A, B**) Determination of the TAX-6/CnA place of action in the control of thermo-nociceptive responses, using cell-specific expression of a TAX-6/CnA gain-of-function mutant. Heat-evoked response in naive animals (T0) and after 60 min of repeated stimulations (T60), scored and reported as in *Figure 2*. The promoters used to drive *tax-6(gf)* cDNA expression are indicated below each bar. **\*\*p < 0.01 and *p < 0.05 versus N2(WT) by Bonferroni–Holm post hoc tests. ns, not significant. *tax-6(gf)* mutant data are shown for comparison purpose. (**C**) Schematics of the hypothetical circuit controlling noxious-heat-evoked reversals as in *Figure 4D*, with the main loci of action of TAX-6/CnA highlighted in red. TAX-6/CnA-evoked up-regulation of thermo-nociceptive response in naive animals (top) and TAX-6/CnA-evoked inhibition of adaptation (bottom).

The online version of this article includes the following source data for figure 5:

**Source data 1.** Numerical data and p values presented in the figures.

Second, regarding adaptation to repeated heat stimuli, we found that the *[glr-1p::tax-6(gf)]* transgene could significantly reduce the adaptation effect, leaving stronger response as compared to wild type after 1 hr of repeated stimulation (*Figure 5A*). This effect was not as strong as the effect in *tax-6(gf)* mutant, which could be due to the transgene overexpression, mosaicism in the transgenic animals or to the fact that TAX-6/CnA overactivation in additional (unidentified) neurons could be also needed to reach the full effect. In contrast, the *[mec-3p::tax-6(gf)]* transgene did not affect nociceptive adaptation, whereas the *[tax-4p::tax-6(gf)]* transgene produced a slight enhancement of adaptation (p < 0.05). The data in *Figure 5A* suggest that one or more *glr-1*-expressing interneurons represent a major place of action in which overactive TAX-6/CnA signaling can act to inhibit adaptation. We deepened the circuit analysis with additional promoters and observed a marked adaptation defect with *cex-1p* and *tdc-1p* promoters, similar to that with *glr-1p* promoter, and a milder defect with *lgc-39p* and *nmr-1p* promoter. These results point to RIM second layer interneurons as primary locus and AVA/AVD/AVE reversal command neurons as secondary locus of overactive TAX-6/CnA action, while not ruling out the implication of additional neurons (*Figure 5C*, bottom).

## Discussion

### CMK-1 substrate specificity is indistinguishable from that of mammalian CaMKs

Our knowledge on CaMK substrate specificity comes from numerous previous biochemical studies on mammalian CaMKs using synthetic peptide substrates at different scales (*Lee et al., 1994*; *White et al., 1998*; *Corcoran et al., 2003*; *Johnson et al., 2023*). Our study complements and extends this previous work through a large-scale identification of CaMKI substrates with endogenous peptide and protein libraries. With a dataset comprising several hundred substrates, we could confirm the $\Phi$XRXX-(S/T)XXX$\Phi$ consensus (where $\Phi$ represents hydrophobic residues) (*Lee et al., 1994*; *White et al., 1998*). These findings suggest that (1) the vast majority of our hits are direct CMK-1 substrates in vitro and (2) the substrate specificity is conserved from worm to mammals.

### Large-scale empirical identification of CMK-1 phospho-targets

We have identified several hundred substrates that can be directly phosphorylated by CMK-1 in vitro. Our hit list is expected to include both substrates that are relevant in vivo and substrates that are not. Indeed, both the peptide library and the whole protein library datasets could include phospho-sites coming from proteins that never encounter CMK-1 in vivo: for example, non-neuronal proteins, secreted proteins, or proteins residing in specific organelles. Furthermore, both datasets are expected to be biased toward abundant proteins, which are more likely to be detected. Accordingly, the strongest GO term enrichments in both datasets related to abundant and ubiquitous cellular components, involved in the translation process and cytoskeleton. Even if we cannot rule out an actual inclination of the CaM kinase pathway to regulate these processes, we suspect that these GO term enrichments rather reflect an analytical bias toward abundant proteins.

The relatively low overlap between the peptide and the protein library datasets can be explained by multiple effects. First, the peptide library dataset is expected to include phosphosites in protein regions that are usually inaccessible to CMK-1 because of protein folding or protein complex formation (increasing the false positive rate in the peptide library). Second, the trypsin-based cleavage prior to CMK-1 exposure in the peptide library protocol might impair proper substrate recognition by the kinase (increasing the false negative rate in the peptide library). Third, we have chosen quite strict criteria and applied them separately to define each hit list. For theoretical reasons, because substrates were presented to the kinase in a more native condition, we tend to give more trust to the protein library dataset, with the subset of targets also found in the peptide library as top hits.

A previous study reported a list of 533 CMK-1 candidate substrates based on in silico predictions (*Ardiel et al., 2018*). We found very little overlap between those predictions and our empirical datasets, with only five phosphosites shared with our peptide library list and one with our protein library list (*Figure 1—figure supplement 1*). This limited overlap could be explained either by a limited exhaustiveness across the three datasets (high false negative rate) or come from limitations affecting in silico predictions. In support of the second explanation, we note that the previously predicted phosphosite list only very partially matches the empirically determined CMK-1 substrate consensus (*Figure 1—figure supplement 1*). Because we lack a solid reference point, it is nevertheless hard to estimate the false positive and false negative rates in our analyses at this stage. Additional experiments will be required to determine if the phosphorylation of these substrates is relevant in vivo, potentially on a case-by-case basis. Eventually, and keeping these limitations in mind, we believe that our large-scale dataset on the substrates which can by phosphorylated by CMK-1 in vitro will represent a useful resource in the search for CMK-1/CaMKI substrates mediating its numerous biological functions in vivo.

### The phosphorylation of TAX-6/CnA by CMK-1/CaMKI

Previous in vitro studies have found that CnA can be phosphorylated by protein kinase C, casein kinase I, casein kinase II, and by CaM kinase II (*Hashimoto et al., 1988*; *Hashimoto and Soderling, 1989*; *Calalb et al., 1990*; *Rusnak and Mertz, 2000*). We are not aware of any study having shown that CnA is a substrate of CaM kinase I. Here, we show that *C. elegans* CaMKI CMK-1 can phosphorylate TAX-6/CnA on S443. This specific phosphorylation event was confirmed in three different types of kinase assays: one using a peptide library, one using a protein library, and one using purified recombinant

TAX-6/CnA as substrate. Ser 443 is located just downstream of the CaM-binding domain in a highly conserved region (*Figure 1D, E*) that was previously found to be phosphorylated by mammalian CaMKII in vitro (*Hashimoto et al., 1988*; *Hashimoto and Soderling, 1989*). Because TAX-6/CnA and CMK-1 seem to mostly work in distinct neuronal cell-type to control nociceptive adaptation, it remains unclear whether TAX-6/CnA phosphorylation by CMK-1 is directly relevant for the control of this process in vivo. Nevertheless, since the expression patterns of CMK-1 and TAX-6/CnA largely overlap and since they are both activated by calcium signaling, we speculate that this phosphorylation event might be relevant to other neuronal regulatory signaling and we suggest it should be considered in future studies addressing the potential cross-talks between the two pathways.

## Complex interaction network and distributed cellular locus of actions for CMK-1 and calcineurin signaling

Our systematic analysis of CMK-1 and calcineurin signaling cross-talks has revealed a relatively complex regulatory network in which the two pathways mostly antagonize each other to regulate adaptation. As mentioned above, adaptation still takes place when CMK-1 and TAX-6/CnA are concomitantly inhibited, which led us to model their action as regulatory events controlling a separate adaptation process (see model in *Figure 3D*). While relatively complex, this model is the simplest we could articulate to explain all the empirically measured interactions (see *Figure 3—figure supplement 1* for a case-by-case illustration of the proposed regulatory network functioning in our different experimental conditions). In this model network, CMK-1 can regulate thermal nociception via three pathways: in the first pathway CMK-1 promotes adaptation by inhibiting TAX-6/CnA, which in turn inhibits adaptation. In the second and third pathways, CMK-1 works independently of TAX-6/CnA to inhibit and promote adaptation, respectively. These two latter pathways can each be gated by TAX-6/CnA. Therefore, both CMK-1 and calcineurin signaling activities may promote or inhibit adaptation and shifting their activity balance could represent a way to achieve nuanced yet robust modulation, integrating past activity with potentially additional cues. This complex regulation scheme would be hard to achieve if the two signaling pathways were working in a single neuronal cell type and it is therefore not surprising that our cell-specific approaches revealed multiple cellular loci of action in the circuit controlling temperature sensation and reversal execution.

In a previous study, CMK-1 was shown to work cell-autonomously in FLP to modulate thermo-nociceptive responses following persistent noxious heat stimulation (*Schild et al., 2014*). FLPs are 'tonic' thermo-sensory neurons, whose activity continuously reflects the current temperature (*Saro et al., 2020*). Here, we show that AFDs, but not FLPs, constitute the main cellular locus of action of CMK-1-dependent plasticity in the case of repeated short-lasting stimulations. Interestingly, AFD are mostly 'phasic' thermo-sensory neurons producing activity peaks in response to thermal changes (*Kimura et al., 2004*; *Clark et al., 2006*; *Hawk et al., 2018*; *Glauser, 2022*). They are thus ideally suited to encode repeated thermal pulses and modulate the heat-evoked reversal circuit. Laser ablation of AFD was previously shown to reduce reversal in response to head-targeted infra-red laser beams (*Liu et al., 2012*). Here, we used thermal stimuli that were diffuse (whole animal exposure) and found that neither the genetic ablation of AFD, nor the cell-specific inhibition of neurotransmission affected the animal's ability to produce heat-evoked reversals. Instead, AFD was essential to favor an experience-dependent reduction in heat-evoked reversal response. CMK-1 was previously shown to mediate both short-term (minute timescale) and long-term (hour timescale) adaptation in AFD intracellular calcium activity that resulted from thermal changes within the 15–25°C innocuous temperature range (*Yu et al., 2014*). Long-term changes involved gene expression changes, whereas the processes mediating short-term adaptation remained elusive (*Yu et al., 2014*). At this stage, we do not know exactly how CMK-1 works in AFD to orchestrate thermo-nociceptive adaptation, but we can envision many possible mechanisms, including qualitative or quantitative modulation of AFD thermo-sensitivity, neuronal excitability, calcium dynamics, neurotransmitter or neuro-modulator release, or even developmental effect affecting AFD synaptic connectivity. Likewise, we do not know the downstream circuit involved. Among different possibilities, the CMK-1 signaling taking place in AFD could affect reversal via AFD's direct interneuron partners AIZ and/or AIY, or engage extra-synaptic communication with neuropeptides. Additional studies will thus be needed to address the downstream processes that are engaged within and downstream of AFD to modulate animal responsiveness to noxious heat stimuli.

We found that calcineurin signaling works at multiple cellular loci to control thermo-nociceptive response. On the one hand, overactive TAX-6 activity in FLP is sufficient to increase reversal response in naive animals. Since FLP activity is known to favor reversal response, we might hypothesize that calcineurin activity could promote FLP activity. On the other hand, overactive TAX-6 activity in RIM or, to a lesser extent, in AVA/AVD/AVE neuron is sufficient to increase reversal response in naive animals, but conversely enhances reversal response upon repeated stimulation. This suggests that the impact that calcineurin signaling has in these neurons depends on past experience. AVA/AVD/AVE are thought to mostly promote reversal response, whereas RIM neuron activity might either up- or down-regulate reversal responses based on the context (*Alkema et al., 2005*; *Gray et al., 2005*; *Guo et al., 2009*; *Piggott et al., 2011*; *Cho and Sternberg, 2014*; *Li et al., 2023*). One particularly interesting observation is that cell-autonomous TAX-6 overactivation in interneurons acts as a major gate blocking adaptation irrespective of any CMK-1 activity manipulation. One hypothetical mechanism could be that TAX-6/CnA activity in worm would work in post-synaptic region of interneuron to adjust synaptic strength, for example, via its action on ion channels or on inhibitory or excitatory neurotransmitter receptors, as shown in various models of synaptic plasticity in mammals (*Groth et al., 2003*).

## Conclusion

In summary, our study reports the empirical identification of many potential CMK-1 targets in vitro, among which the catalytic subunit of Calcineurin TAX-6/CnA. Whereas we have not yet found evidence that the phosphorylation of TAX-6 by CMK-1 is directly relevant for thermo-nociceptive response modulation, we show a complex interplay between CMK-1 and calcineurin signaling that operate via multiple regulatory nodes within the noxious-heat-evoked reversal circuits of *C. elegans*. Our study paves the way for a deeper dissection of how conserved intracellular signaling pathways operating at distributed loci within a sensory-behavior circuit can actuate experience-dependent changes in nociceptive behavior.

# Materials and methods
## Worm maintenance

All *C. elegans* strains used in this study are listed in *Supplementary file 3*. All strains were grown on nematode growth media (NGM) plates with OP50 *E. coli* (Stiernagle, 2006) at 20°C. For TAX-6 inhibition experiments, NGM plates containing cyclosporine A were used, as well as regular NGM plates as control. Cyclosporine A (10 µM) plates were prepared 72 hr before experiments and kept protected from light at room temperature (RT).

## Expression and purification of CMK-1(T179D)-GST, CMK-1(K52A)-GST, and TAX-6-HIS6

DNA fragments encoding CMK-1 and TAX-6 proteins were PCR amplified and cloned into NdeI and BamHI restriction sites of pDK2409 (pET-24d/GST-TEV-KAP104.419C) for the GST-TEV-tagged protein and of pDK2832 (pET-24d/(His)6) for the His6-tagged proteins. Plasmids were transformed into *E. coli* BL21 (DE3) (Novagen). For protein expression, bacteria were grown overnight at 37°C, next day transferred to 200 ml of Luria Broth (LB) with kanamycin 30 µg/ml to $OD_{600}$ = 0.1, incubated up to $OD_{600}$ = 0.5. Then, protein expression was induced using 0.5 mM isopropyl β-D-1-thiogalactopyranoside and incubated for 5 hr at 23°C. After, bacteria cells were collected by centrifugation at 2800 rcf for 10 min at 4°C, resuspended in cold lysis buffer (150 mM NaCl, 50 mM Tris-HCl pH7.5, 1.5 mM $MgCl_2$, 5% glycerol, 1 mM phenylmethylsulfonyl fluoride (PMSF)) and lysed in a Microfluidizer Processor M-110L. Then, NP-40 was added to the concentration of 0.1%, and soluble extract was obtained by centrifugation at 23,400 rcf for 20 min at 4°C. Supernatant was incubated for 2 hr at 4°C while rotating with Glutathione superflow beads (QIAGEN) for GST-tagged CMK-1 variants or nickel-nitrilotriacetic acid beads (Ni-NTA, QIAGEN) in the presence of 15 mM imidazole for TAX-6-His6 protein binding. Beads were harvested at 580 rcf at 4°C. GST-tagged CMK-1 kinase was cleaved from the tag during incubation in lysis buffer (added 1 mM PMSF, 0.1% NP-40, 1 mM dithiothreitol (DTT)) with home-made TEV protease. TAX-6-bound Ni-NTA beads were washed seven times with imidazole-containing buffer (5× with 15 mM and 2× with 50 mM imidazole) and eluted in lysis buffer with 1 mM PMSF, 0.1% NP-40,

and 500 mM imidazole. Protein concentration was determined by Pierce Microplate BCA protein assay Kit-Reducing Agent Compatible (Thermo Scientific) using BSA as protein standard.

## Total protein lysate preparation

Worms were grown on 10 cm diameter plates to the stage of young adults, washed with distilled water and suspended in extraction buffer. For in vitro kinase assays on peptide, urea buffer (8 M urea, 50 mM Tris-HCl pH 8.5) was used. For in vitro kinase assays on intact proteins, native protein extraction buffer (50 mM HEPES pH 7.4, 1% NP-40, 150 mM NaCl, 1× protease inhibitors, 0.5 mM PMSF) was used. Worms were flash-frozen in liquid nitrogen and cryogenically disrupted by using a Precellys homogenizer and acid-washed glass beads (5000 × 3 for 30 s with 30 s pause after each cycle). Lysates were collected by centrifugation at 1500 rcf at 4°C.

## In vitro kinase assay

Purified CMK-1 kinase was used the same day to avoid freezing. In vitro kinase assays were performed according to *Hu et al., 2021*. Briefly, for kinase assays performed on native proteins, 30 mg of worm protein extract was incubated with pre-washed NHS-activated Sepharose beads at 4°C on a rotor for 6 hr. Beads were washed with 3 × 10 ml of phosphatase buffer (50 mM HEPES, 100 mM NaCl, 0.1% NP-40). 1 ml of phosphatase buffer containing 5000–10,000 units of lambda phosphatase with 1 mM $MnCl_2$ was added and incubated for 4 hr at RT followed by overnight incubation at 4°C on the rotor to dephosphorylate endogenous proteins. Beads were washed with 2 × 10 ml of kinase buffer (50 mM Tris-HCl pH 7.6, 10 mM $MgCl_2$, 150 mM NaCl and 1× PhosSTOP (Roche)). Endogenous kinases bound to beads were inhibited by incubation with 1 mM FSBA in 1 ml of kinase buffer at RT on the rotor for 2 hr. In addition, inhibition of the remaining active kinases was achieved with a further 1 hr incubation in the presence of staurosporine (LC Laboratories), added to a final concentration of 100 µM. The beads were washed with 3 × 10 ml of kinase buffer to remove non-bound kinase inhibitors. The supernatant was removed completely using gel loading tips. Beads were split into six tubes 3× with kinase and 1 mM ATP, 3× with kinase and without ATP (Sigma-Aldrich), kinase buffer was added and incubated for 4 hr at 30°C while shaking. For TAX-6 phosphorylation by CMK-1 in vitro assay, CMK-1 was kept on Glutathione superflow beads and purified TAX-6 protein in kinase buffer (50 mM HEPES pH7.5, 10 mM Mg(Ac)$_2$, 1 mM DTT, 1× PhosSTOP) was added onto protein/bead mix together with ATP. Afterwards samples were lyophilized followed by protein digestion using trypsin. Samples were chemically labeled using dimethyl-labeling (*Boersema et al., 2009*) supporting relative MS-based quantification. For kinase assays performed on peptide libraries, the procedure was the same except that protein extract samples were digested with Lyc-C (Lysyl Endopeptidase, WAKO) for 4 hr and trypsin (Promega) overnight, before incubating them with Sepharose beads. Later beads were incubated either with purified active CMK-1 or kinase dead CMK-1 in kinase buffer (50 mM Tris-HCl pH 7.6, 10 mM $MgCl_2$, 150 mM NaCl, 1× PhosSTOP, 1 mM DTT, and 1 mM γ-[$^{18}$O4]-ATP (Cambridge Isotope Laboratories)).

Peptides were fractionated and phosphopeptides enriched as described (*Hu et al., 2021*); briefly: samples were incubated with TiO2 (GL Sciences) slurry, which was pre-incubated with 300 mg/ml lactic acid in 80% acetonitrile, 1% trifluoroacetic acid (TFA) prior to enrichment for 30 min at RT. For elution, $TiO_2$ beads were transferred to 200 µl pipette tips blocked by C8 discs. After washing with 10% acetonitrile/1% TFA, 80% acetonitrile/1% TFA, and LC–MS grade water, phosphopeptides were eluted with 1.25% ammonia in 20% acetonitrile and 1.25% ammonia in 80% acetonitrile. Eluates were acidified with formic acid, concentrated by vacuum concentration, and resuspended in 0.1% formic acid for LC–MS/MS analysis.

## LC–MS/MS analyses

LC–MS/MS measurements were performed on two LC–MS/MS systems as described (*Hu et al., 2021*): a QExactive (QE) Plus and HF-X mass spectrometer coupled to an EasyLC 1000 and EasyLC 1200 nanoflow-HPLC, respectively (all Thermo Scientific). Peptides were fractionated on fused silica HPLC-column tips using a gradient of A (0.1% formic acid in water) and B (0.1% formic acid in 80% acetonitrile in water): 5–30% B within 85 min with a flow rate of 250 nl/min. Mass spectrometers were operated in the data-dependent mode; after each MS scan ($m/z$ = 370–1750; resolution: 70,000 for QE Plus and 120'000 for HF-X) a maximum of ten, or twelve MS/MS scans were performed (normalized

collision energy of 25%), a target value of 1000 (QE Plus)/5000 (HF-X) and a resolution of 17,500 for QE Plus and 30,000 for HF-X. MS raw files were analyzed using MaxQuant (version 1.6.2.10) (*Cox and Mann, 2008*) using a full-length *C. elegans* Uniprot database (November 2017), and common contaminants such as keratins and enzymes used for in-gel digestion as reference. Carbamidomethyl-cysteine was set as fixed modification and protein amino-terminal acetylation, serine-, threonine- and tyrosine- (heavy) phosphorylation, and oxidation of methionine were set as variable modifications. In case of labeled samples, triple dimethyl-labeling was chosen as quantification method (*Boersema et al., 2009*). The MS/MS tolerance was set to 20 ppm and three missed cleavages were allowed using trypsin/P as enzyme specificity. Peptide, site, and protein FDR based on a forward-reverse database were set to 0.01, minimum peptide length was set to 7, the minimum score for modified peptides was 40, and minimum number of peptides for identification of proteins was set to one, which must be unique. The 'match-between-run' option was used with a time window of 0.7 min. MaxQuant results were analyzed using Perseus (*Tyanova et al., 2016*).

## Worm preparation for heat avoidance assay and number of replicates

Gravid adult worms were treated with hypochlorite solution according to standard protocol. Embryos were rinsed twice with water and once with M9 buffer, then resuspended in M9 and plated on NGM plates seeded with OP-50 *E. coli*. 200–300 embryos per individual plate were seeded. Worms were incubated at 20°C until start of egg laying (65–90 hr, depending on the strain or condition). Worms were washed off the plates with distilled water, placed into 1.5 ml tubes and washed twice more to remove bacterial residues. Worms were placed on unseeded NGM plates and left to disperse and acclimate in the experimental room for 60 min. Prior to worm deposition, these experimental unseeded plates were kept open in a laminar flow hood for 3 hr to ensure dry surface. Plate lid was removed 3 min before starting the assay. At least three replicates were performed for each strain or condition on three separate days running wild-type N2 strain alongside.

For the experiments using transgenic animals, fluorescent signal-carrying worms were picked 16–19 hr before the experiment to allow for recovery.

## Heat stimulation and adaptation protocols

For the majority of experiments, we compared the heat-evoked response in naive animals (that had never been stimulated, T0) and the same animals after 60 min of repeated stimulation (4 s stimuli and 20 s ISI, T60). For some experiments, the adaptation period was reduced to 10 min (T10) or 30 min (T30). The INFERNO system (*Lia and Glauser, 2020*) was used for behavioral recordings. The heat stimulation program during recordings was composed of a 40-s baseline period without any heat stimulation, a 4-s stimulation with 400 W heating (4 IR lamps turned on) and a 20-s post-stimulation period. The repeated stimulation delivery (adaptation treatment) was achieved by placing the worm plates under the ThermINATOR system (*Lia and Glauser, 2020*), providing infinitely looping temperature program composed of 4 s of IR lamps stimulation, followed by 20 s ISI with the lamps turned off. There was a lag of about 10 s for the plate transfer between the two systems.

In the INFERNO system, worm plates were recorded using a DMK 33U×250 camera and movies acquired with the IC capture software (The Imaging Source), at 8 frames per second, at a 1600 × 1800 pixel resolution, and the resulting .AVI file was encoded as Y800 8-bit monochrome. The Multi-Worm Tracker 1.3.0 (MWT) (*Swierczek et al., 2011*) with previously described configuration settings (*Lia and Glauser, 2020*) was used for movie analyses. A previously described Python script was use to flag the frame of reversal occurrence (*Lia and Glauser, 2020*). Each reported data point corresponds to the results of one assay plate scoring at least 50 animals.

## Transgene construction and transgenesis

Promoter-containing Entry plasmids (Multi-site Gateway slot 1) were constructed by PCR using N2 genomic DNA as template and primers flanked by attB4 and attB1r recombination sites; the PCR product being cloned into pDONR-P4-P1R vector (Invitrogen) by BP recombination.

Coding sequence-containing Entry plasmids (Multi-site Gateway slot 2) were constructed by PCR using N2 cDNA as template and primers flanked by attB1 and attB2 recombination sites; the PCR product being cloned into pDONR_221 vector (Invitrogen) by BP recombination.

Expression plasmids for transgenesis were created through LR recombination reactions (Gateway LR Clonase, Invitrogen) as per the manufacturer's instructions.

Primer sequences for BP reactions and all plasmids used in this study are listed in *Supplementary file 3*.

DNA constructs were transformed into competent DH5a *E. coli* (NEB C2987H), purified with the GenElute HP Plasmid miniprep kit (Sigma) and microinjected in the worm gonad according to a standard protocol (*Evans, 2006*) together with co-injection markers for transgenic animal identification. The concentrations of expression plasmids and co-markers injected are indicated in *Supplementary file 3*.

### Site-directed mutagenesis

To create truncated *tax-6(gf)* transgene PCR-based site-directed mutagenesis (*Hemsley et al., 1989*) was used. Entry plasmid containing *tax-6* coding DNA sequence was amplified with the KOD Hot Start DNA Polymerase (Novagen; Merck). Primers were designed to contain the desired deletion and phosphorylated in 5′ end (*Supplementary file 3*). Linear PCR products were purified from agarose gel (1%) after electrophoresis with a Zymoclean-Gel DNA Recovery kit (Zymo Research), circularized using DNA Ligation Kit <Mighty Mix> (Takara).

### Statistical analyses

ANOVAs were conducted using Jamovi (The jamovi project 2022), jamovi (Version 2.3) [Computer Software]; retrieved from https://www.jamovi.org. Post hoc tests were used to compare each of the mutants with wild type (N2) or *cmk-1(lf)* using Bonferroni–Holm correction.

### Acknowledgements

We are grateful to Lisa Schild, Laurence Bulliard, and Michael Stumpe for expert technical support. Some strains were provided by the CGC, which is funded by NIH Office of Research Infrastructure Programs (P40 OD010440). The study was supported by the Swiss National Science Foundation (BSSGI0_155764, PP00P3_150681, and 310030_197607 to DAG, 310030_212187 to JD) and by the Novartis Foundation for Medical-Biological Research.

## Additional information

### Funding

| Funder | Grant reference number | Author |
|---|---|---|
| Schweizerischer Nationalfonds zur Förderung der Wissenschaftlichen Forschung | BSSGI0_155764 | Dominique A Glauser |
| Schweizerischer Nationalfonds zur Förderung der Wissenschaftlichen Forschung | PP00P3_150681 | Dominique A Glauser |
| Schweizerischer Nationalfonds zur Förderung der Wissenschaftlichen Forschung | 310030_197607 | Dominique A Glauser |
| Schweizerischer Nationalfonds zur Förderung der Wissenschaftlichen Forschung | 310030_212187 | Jörn Dengjel |

| Funder | Grant reference number | Author |
|---|---|---|
| Novartis Stiftung für Medizinisch-Biologische Forschung | | Dominique A Glauser |

The funders had no role in study design, data collection, and interpretation, or the decision to submit the work for publication.

## Author contributions

Martina Rudgalvyte, Conceptualization, Formal analysis, Investigation, Visualization, Methodology, Writing - original draft, Writing - review and editing; Zehan Hu, Formal analysis, Investigation, Methodology; Dieter Kressler, Conceptualization, Formal analysis, Methodology, Project administration; Jörn Dengjel, Conceptualization, Formal analysis, Supervision, Methodology, Project administration, Writing - review and editing; Dominique A Glauser, Conceptualization, Formal analysis, Supervision, Funding acquisition, Visualization, Writing - original draft, Project administration, Writing - review and editing

## Author ORCIDs
Martina Rudgalvyte (ID) https://orcid.org/0009-0003-6201-6200
Dieter Kressler (ID) https://orcid.org/0000-0003-4855-3563
Jörn Dengjel (ID) https://orcid.org/0000-0002-9453-4614
Dominique A Glauser (ID) https://orcid.org/0000-0002-3228-7304

Reviewer #1 (Public review): https://doi.org/10.7554/eLife.103497.3.sa1
Reviewer #2 (Public review): https://doi.org/10.7554/eLife.103497.3.sa2
Author response https://doi.org/10.7554/eLife.103497.3.sa3

# Additional files

## Supplementary files
Supplementary file 1. List of phophosites identified in the different CMK-1 in vitro kinase assays.

Supplementary file 2. Gene ontology (GO) terms enriched in the CMK-1 target subsets.

Supplementary file 3. Strain and plasmid lists.

MDAR checklist

## Data availability
MS data have been deposited to the ProteomeXchange Consortium via the PRIDE partner repository with the dataset identifier PXD055776 (*Perez-Riverol et al., 2022*). The numerical data and p values presented in Figures 2–5 are provided.

The following dataset was generated:

| Author(s) | Year | Dataset title | Dataset URL | Database and Identifier |
|---|---|---|---|---|
| Dengjel J | 2025 | Antagonist actions of CMK-1/CaMKI and TAX-6/Calcineurin along the *C. elegans* thermal avoidance circuit orchestrate nociceptive habituation | https://www.ebi.ac.uk/pride/archive/projects/PXD055776 | PRIDE, PXD055776 |

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
