## [Editor Report · eLife Assessment]

This study uses *C. elegans* to investigate how the Calcium/Calmodulin-dependent kinase CMK-1 regulates adaptation to thermo-nociceptive stimuli. The authors use **compelling** approaches to identify Calcineurin as a phosphorylation target of CMK-1 and to investigate the relationship between CMK-1 and Calcineurin using gain and loss of function genetic and pharmacological methods. The findings of this study are **valuable** as they show that CMK-1 and Calcineurin act in separate neurons in an antagonistic and complex manner to regulate thermo-nociceptive adaptation, and these results may be relevant for understanding some chronic human pain conditions.

---

## [Referee Report · Reviewer #1 (Public review)]

Summary:

Goal: Find downstream targets of cmk-1 phosphorylation, identify one that also seems to act in thermosensory habituation, test for genetic interactions between cmk-1 and this gene and assess where these genes are acting in the thermosensory circuit during thermosensory habituation.

Methods: Two in vitro analyses of cmk-1 phosphorylation of *C. elegans* proteins. Thermosensory habituation of cmk-1 and tax-6 mutants and double mutants was assessed by measuring rate of heat evoked reversals (reversal probability) of *C. elegans* before and after 20s ISI repeated heat pulses over 60 minutes.

Conclusions: cmk-1 and tax-6 act in separate habituation processes primarily in AFD, that interact complexly, but both serve to habituate the thermosensory reversal response. They found that cmk-1 primarily acts in AFD and tax-6 primarily acts in RIM (and FLP for naïve responses). They also identified hundreds of potential cmk-1 phosphorylation substrates in vitro.

Strengths:

The effects size in the genetic data is quite strong and a large number of genetic interaction experiments between cmk-1 and tax-1 demonstrate a complex interaction.

A major concern concerning this manuscript was the assumption that the process they are observing is habituation. The two previously cited papers using this (or a very similar) protocol, Lia and Glauser 2020 and Jordan and Glauser 2023, both use the word 'adaptation' to describe the observed behavioral decrement. Jordan and Glauser 2023 does occasionally use the words 'habituation' or 'habituation-like' 10 times, however it uses 'adaptation' over 100 times. It is critical to distinguish habituation from sensory adaptation (or fatigue) in this thermal reversal protocol. These processes are often confused/conflated, however they are very different; sensory adaptation is a process that decreases how much the nervous system is activated by a repeated stimulus, therefore it can even occur outside of the nervous system. Habituation is a learning process where the nervous system responds less to a repeated stimulus, despite (at least part of the nervous system) the nervous system still being similarly activated by the stimulus. Habituation is considered an attentional process, while adaptation is due to fatigue of sensory transduction machinery. Control experiments such as tests for dishabituation (where application of a different stimulus causes recovery of the decremented response) or rate of spontaneous recovery (more rapid recovery after short inter-stimulus intervals) are required to determine if habituation or sensory adaptation are occurring. These experiments will allow the results to be interpreted with clarity; without them, it isn't actually clear what biological process is actually being studied. The authors have accepted this distinction and now correctly call the process adaptation.

While there was originally some discrepancy between the two in vitro phosphorylation experiments and the in silico predictions, the revision has cleared up the issues.

Figure 3 -S1: This model has been adjusted to more closely fit the data.

The authors have expanded the discussion about the significance of the sites of cmk-1 and tax-6 function in the neural circuit.

---

## [Referee Report · Reviewer #2 (Public review)]

Summary:

The reduction in a response to a specific stimuli after repeated exposures is called habituation. Alterations in habituation to noxious stimuli are associated with chronic pain in humans, however the underlying molecular mechanisms involved are not clear. This study uses the nematode *C. elegans* to study genes and mechanisms that underlie adaptation to a form of noxious stimuli based on heat, termed thermo-noxious stimuli. The authors previously showed that the Calcium/Calmodulin-dependent protein kinase (CMK-1) regulates thermo-nociceptive adaptation in the nematode *C. elegans*. Although CMK-1 is a kinase with many known substrates, the downstream targets relevant for thermo-nociceptive adaptation are not known. In this study, the authors use two different kinase screens to identify phosphorylation targets of CMK-1. One of the targets they identify is Calcineurin (TAX-6). The authors show that CMK-1 phosphorylates a regulatory domain of Calcineurin at a highly conserved site (S443). In a series of elegant experiments, the authors use genetic and pharmacological approaches to increase or decrease CMK-1 and Calcineurin signaling to study their effects on thermo-nociceptive adaptation in *C. elegans*. They also combine these various approaches to study the interactions between these two signaling proteins. The authors use specific promoters to determine in which neurons CMK-1 and Calcineurin function to regulate thermo-nociceptive adaptation. The authors propose a model based on their findings, illustrating that CMK-1 and Calcineurin act mostly in different neurons to antagonistically regulate adaptation to thermo-nociceptive stimuli in a complex manner.

Strengths:

- Given the conservation of adaptation across phylogeny, identifying genes and mechanisms that underlie nociceptive adaptation in *C. elegans* may be relevant for understanding chronic pain in humans.

- The identification of canonical CaM Kinase phosphorylation motifs in the substrates identified in the CMK-1 substrate screen validates the screen.

- The use of loss and gain of function approaches to study the effects of CMK-1 and Calcineurin on thermo-nociceptive responses and adaptation is elegant.

- The ability to determine the cellular place of action of CMK-1 and Calcineurin using neuron specific promoters in the nematode is a clear strength of the genetic model system.

Weaknesses:

- The manuscript begins by identifying Calcineurin as a direct substrate of CMK-1 but ends by showing that CMK-1 and Calcineurin mostly act in different neurons to regulate nociceptive adaptation, thus the physiological relevance of CMK-1 phosphorylation of Calcineurin is not clear.

---

## [Author Response]

The following is the authors’ response to the original reviews

**Public Reviews:**

**Reviewer #1 (Public review):**
Summary:Goal: Find downstream targets of cmk-1 phosphorylation, identify one that also seems to act in thermosensory habituation, test for genetic interactions between cmk-1 and this gene, and assess where these genes are acting in the thermosensory circuit during thermosensory habituation.Methods: Two in vitro analyses of cmk-1 phosphorylation of *C. elegans* proteins. Thermosensory habituation of cmk-1 and tax-6 mutants and double mutants was assessed by measuring the rate of heat-evoked reversals (reversal probability) of *C. elegans* before and after 20s ISI repeated heat pulses over 60 minutes.Conclusions: cmk-1 and tax-6 act in separate habituation processes, primarily in AFD, that interact complexly, but both serve to habituate the thermosensory reversal response. They found that cmk-1 primarily acts in AFD and tax-6 primarily acts in RIM (and FLP for naïve responses). They also identified hundreds of potential cmk-1 phosphorylation substrates in vitro.Strengths:The effect size in the genetic data is quite strong and a large number of genetic interaction experiments between cmk-1 and tax-1 demonstrate a complex interaction.

Thanks a lot for these positive remarks.

Weaknesses:The major concern about this manuscript is the assumption that the process they are observing is habituation. The two previously cited papers using this (or a very similar) protocol, Lia and Glauser 2020 and Jordan and Glauser 2023, both use the word 'adaptation' to describe the observed behavioral decrement. Jordan and Glauser 2023 use the words 'habituation' or 'habituation-like' 10 times, however, they use 'adaptation' over 100 times. It is critical to distinguish habituation from sensory adaptation (or fatigue) in this thermal reversal protocol. These processes are often confused/conflated, however, they are very different; sensory adaptation is a process that decreases how much the nervous system is activated by a repeated stimulus, therefore it can even occur outside of the nervous system. Habituation is a learning process where the nervous system responds less to a repeated stimulus, despite (at least part of the nervous system) the nervous system still being similarly activated by the stimulus. Habituation is considered an attentional process, while adaptation is due to the fatigue of sensory transduction machinery. Control experiments such as tests for dishabituation (where the application of a different stimulus causes recovery of the decremented response) or rate of spontaneous recovery (more rapid recovery after short inter-stimulus intervals) are required to determine if habituation or sensory adaptation are occurring. These experiments will allow the results to be interpreted with clarity, without them, it isn't actually clear what biological process is actually being studied.

Thanks for the comment. As this reviewer points out, “adaptation” and “habituation” are often conflated. Many scientists (maybe not the majority though) use a less stringent definition for the word habituation, than the one presented by this reviewer. More particularly, the term habituation is used in human pain research to refer solely to the reduction of response to repeated stimuli, in the absence of a detailed assessment of the more stringent criteria mentioned here (see, e.g., PMID: 22337205 ; PMID: 18947923 ; PMID: 17258858; PMID: 20685171 ; PMID: 15978487). In addition to the practice in pain research, the main reason why we steered toward ‘habituation’ from our previous publication is because it immediately conveys the idea of a response reduction, whereas ‘adaptation’ could in principle be either an up-regulation or a downregulation of the response (again, based on various definitions). But we agree that using the word “habituation” came at the cost of triggering a confusion about the exact nature of the process, for those considering the stricter definition of the word “habituation” and those not in the narrower field of pain research. In the revised manuscript, we have thus changed this terminology to “adaptation”. Also following suggestions from Reviewer 2, we have strengthened the description of the protocol in the Result section and clarified, why the adaptation phenomenon is not a ‘thermal damage’ effect or ‘fatigue’ effect in the neuro-muscular circuit controlling reversal. One of the most convincing piece of evidence it cannot be solely explained by “damages” or “exhaustion” is simply the existence of non-adapting mutants (like *cmk-1(lf)*) or pharmacological treatments (Cyclosporin A) blocking the adaptation effect and enabling worm to continuously reverse for hours without any problems.

While the discrepancy between the in vitro phosphorylation experiments and the in silico predictions was discussed, the substantial discrepancy (over 85% of the substrates in the smaller in vitro dataset were not identified in the larger dataset) between the two different in vitro datasets was not discussed. This is surprising, as these approaches were quite similar, and it may indicate a measure of unreliability in the in vitro datasets (or high false negative rates).

Thanks for the comment. This is an important aspect which we now more extensively cover in the Discussion section.

The strong consistency of the CMK-1 recognition consensus sequences across the two in vitro dataset speaks against the unreliability of the analyses. Instead, there are a few points to highlight that explain the somewhat low degree of overlap between the two datasets, which indeed relate to the false negative rates as this reviewer suggests.

(1) In the peptide library analysis, Trypsin cleavage prior to kinase treatment will leave a charged N-term or C- terminus and in addition remove part of the protein context required for efficient kinase recognition. This will have a variable effect across the different substrates in the peptide library, depending on the distance between the cleavage site and the phosphosite, but will not affect the native protein library. This effect increases the false negative rate in the peptide library.

(2) The number and distribution of “available substrate phosphosites” diverge in the two libraries. Indeed, the peptide library is expected to contain a markedly larger diversity of potential CMK-1 substrate sites than the protein library (because the Trypsin digestion will reveal substrates that are normally buried in a native protein), but the depth of MS analysis is the same for the two libraries. In somewhat simplistic terms, the peptide-library analysis is prone to be saturated with abundant phosphorylated peptides, which prevent detecting all phosphosites. If the peptide analysis could have been made deeper, we would probably have increased the overlap (at the cost of increasing the number of false positive too).

(3) We have chosen quite strict criteria and applied them separately to define each hit list; therefore, we know we have many false negatives in each list, which will naturally reduce the expected overlap.

We now extended the discussion of the limited overlap of the two dataset in a dedicated paragraph in the discussion. We also clarify that we tend to give more trust to the protein-library dataset (since substrates are in a configuration closer to that in vivo), with those hits also present in the peptide dataset (like TAX-6 was) as the most convincing hits, as they could be validated in a second type of experiment.

Additionally, the rationale for, and distinction between, the two separate in vitro experiments is not made clear.

We reasoned that both substrate types have their own benefits and limitations (as discussed in the manuscript), so it was an added value to run both. We proposed that the subset of targets present in both datasets to be the most solid list of candidates. We have reinforced this point in the discussion.

Line 207: After reporting that both tax-6 and cnb-1 mutants have high spontaneous reversals, it is not made clear why cnb-1 is not further explored in the paper. Additionally, this spontaneous reversal data should be in a supplementary figure.

We kept the focus of the article primarily on TAX-6, because it was identified as CMK-1 target in vitro; CNB-1 was not. Moreover, we didn’t have *cnb-1(gf)* mutants to pursue the analysis with, and we were stuck by the *cnb-1(lf)* constitutive high reversal rate for any further follow up. We have added a supplementary file to present the spontaneous reversals rates.

Figure 3 -S1: This model doesn't explain why the cmk-1(gf) group and the cmk-1(gf) +cyclo A group cause enhanced response decrement (presumably by reducing the inhibition by tax-6) but the +cyclo A group (inhibited tax-6) showed weaker response decrement, as here there is even further weakened inhibition of tax-6 on this process. Also, the cmk-1(lf) +cyclo A group is labeled as constitutive habituation, however, this doesn't appear to be the case in Figure 3 (seems like a similar initial level and response decrement phenotype to wildtype).

Thanks a lot for the comment. We are glad that the presentation of our complex dataset was clear enough to bring the reader to that level of detailed reflection and interpretation on the proposed model. To address the two points raised in this reviewer comment, we made modifications to the model presentation and provide additional clarifications below, where we use the term adaptation instead of habituation (as in the revised Figure):

Regarding the first point, “why the cmk-1(gf) group and the cmk-1(gf) +cyclo A group cause enhanced response decrement … but the +cyclo A group showed weaker response decrement”. This is really a very good point, that cannot be easily explained if all the branches (arrows) in the model have the same weight or work as ON/OFF switches. We tried to convey the relative importance of the regulation effect via the thickness of the arrow lines (which we have now clarified in the legend in the revised ms). The main ‘quantitative’ nuances to take into consideration here originate from 2 assumptions of the model (which we have clarified in the revised ms):

Assumption 1: the inhibitory effect of TAX-6 on the CMK-1 antiadaptation branch and the inhibitory effect of TAX-6 on the CMK-1 pro-adaptation branch are not of the same magnitude (we have further enhanced the line thickness differences in the revised model, top left panel for wild type).

Assumption 2: the two antagonistic direct effects of CMK-1 on adaptation are not of the same magnitude, most strikingly in the context of CMK-1(gf) mutants.

In our model, the cyclosporin A treatment alone (bottom left panel) causes a strong boost on the CMK-1 inhibitory branch and a less marked boost on the CMK-1 activator branch (following assumption 1). This causes an imbalance between the two antagonist direct CMK-1-dependent drives, which reduces (but doesn’t fully block) adaptation. Indeed, we don’t observe a total block of adaptation with cyclosporin A in wild type, the effect being significantly milder than the totally nonadapting phenotypes seen, e.g., in TAX-6(gf) mutants. From there, the question is what happen in CMK-1(gf) background that would mask the anti-adaptation effect of Cyclosporin A? Here assumption 2 is relevant, and the CMK-1(gf) pro-adaptation direct branch is always prevalent and imbalances the regulation toward faster adaptation (the role of TAX-6 becoming negligible in the CMK-1(gf) background and *ipso facto* that of Cyclosporin A).

Regarding the second point, “the cmk-1(lf) +cyclo A group is labeled as constitutive habituation”. We regret a confusing word choice in the first version of the manuscript; we intended to mean “normal habituation phenotype” but in the joint absence of antagonistic CMK-1 and TAX-6 regulatory signaling (so the regulation is not like in wild-type, but the phenotype ends up like in wild type). We have modified the label to “normal adaptation” and left a note in the legend that an apparently normal adaptation phenotype seems to be the default situation when the two antagonistic regulatory pathways are shut off.

More discussion of the significance of the sites of cmk-1 and tax-6 function in the neural circuit should take place. Additionally, incorporating the suspected loci of cmk-1 and tax-6 in the neural circuit into the model would be interesting (using proper hypothetical language). For example, as it seems like AFD is not required for the naïve reversal response but just its reduction, cmk-1 activity in AFD might be generating inhibition of the reversal response by AFD. It certainly would be understandable if this isn't workable, given extrasynaptic signaling and other unknowns, but it potentially could also be helpful in generating a working model for these complex interactions. For example, cmk1 induces AIZ inhibition of AVA (AIZ is electrically coupled to AFD), and tax-6 reduces RIM activation of AVA (these neurons are also electrically coupled according to the diagram). RIM is also a neuropeptide-rich neuron, so this could allow it to interact with the cmk-1-related process(es) in AFD. Some discussion of possibilities like this could be informative.

Thanks for the comment. These hypothetical inter-cellular communication pathways are indeed nice possibilities. On the other hand, we could envision several additional pathways. While RIM is indeed a neuropeptide-rich neurons, all these neurons actually express neuropeptides. Following this helpful suggestion, we have slightly expanded the discussion of hypothetical cellular pathways that can be modulated downstream of CMK-1 in AFD. We also slightly lengthened the discussion to mention hypothetical post-synaptic target of TAX-6 within interneurons based on the literature.

Provide an explanation for why some of the experiments in Figure 4 have such a high N, compared to other experiments.

The conditions with the highest n correspond to conditions which we have also used as ‘control’ condition for other type of experiments in the lab and as part of side projects, but which could be gathered for the present article. We have been working with *cmk-1(lf)* and *tax-6(gf)* mutants for many years… and the robust non-adapting phenotype was a reference point and a quality control when analyzing other nonadapting mutants.

Because the loss of function and gain of function mutations in cmk-1 have a similar effect, it is likely that this thermosensory plasticity phenotype is sensitive to levels of cmk-1 activity. Therefore, it is not surprising that the cmk-1 promoter failed to rescue very well as these plasmid-driven rescues often result in overexpression. Given this and that the cmk-1p rescue itself was so modest, these rescue experiments are not entirely convincing (and very hard to interpret; for example, is the AFD rescue or the ASER rescue more complete? The ASER one is actually closer to the cmk-1p rescue). Given the sensitivity to cmk-1 activity levels, a degradation strategy would be more likely to deliver clear results (or perhaps even the overactivation approach used for tax-6).

Thanks for the comment. We respectfully disagree with this reviewer’s statement “*the loss of function and gain of function mutations in cmk-1 have a similar effect*”. We suspect a confusion here, because our data clearly show that these two mutant types have an opposite phenotype. That being said, we interpret the weak rescue effect with *cmk-1p* as a probable result of overexpression or incomplete/imbalanced expression across neurons (as the promoter used might not include all the relevant regulatory regions). We dedicated considerable efforts to establish an endogenous CMK-1::degron knock in, for tissue-specific auxin-induced degradation (AID), but we were unfortunately not able to obtain consistent results. Unfortunately, the only useful data regarding CMK-1 place-of-action are the cell-specific rescue data already included in the report.

**Reviewer #2 (Public review):**
Summary:The reduction in a response to a specific stimulus after repeated exposures is called habituation. Alterations in habituation to noxious stimuli are associated with chronic pain in humans, however, the underlying molecular mechanisms involved are not clear. This study uses the nematode *C. elegans* to study genes and mechanisms that underlie habituation to a form of noxious stimuli based on heat, termed thermo-noxious stimuli. The authors previously showed that the Calcium/Calmodulin-dependent protein kinase (CMK-1) regulates thermo-nociceptive habituation in the nematode *C. elegans*. Although CMK-1 is a kinase with many known substrates, the downstream targets relevant for thermo-nociceptive habituation are not known. In this study, the authors use two different kinase screens to identify phosphorylation targets of CMK-1. One of the targets they identify is Calcineurin (TAX-6). The authors show that CMK-1 phosphorylates a regulatory domain of Calcineurin at a highly conserved site (S443). In a series of elegant experiments, the authors use genetic and pharmacological approaches to increase or decrease CMK-1 and Calcineurin signaling to study their effects on thermo-nociceptive habituation in *C. elegans*. They also combine these various approaches to study the interactions between these two signaling proteins. The authors use specific promoters to determine in which neurons CMK-1 and Calcineurin function to regulate thermonociceptive habituation. The authors propose a model based on their findings illustrating that CMK-1 and Calcineurin act mostly in different neurons to antagonistically regulate habituation to thermo-nociceptive stimuli in a complex manner.Strengths:(1) Given the conservation of habituation across phylogeny, identifying genes and mechanisms that underlie nociceptive habituation in *C. elegans* may be relevant for understanding chronic pain in humans.(2) The identification of canonical CaM Kinase phosphorylation motifs in the substrates identified in the CMK-1 substrate screen validates the screen.(3) The use of loss and gain of function approaches to study the effects of CMK-1 and Calcineurin on thermo-nociceptive responses and habituation is elegant.(4) The ability to determine the cellular place of action of CMK-1 and Calcineurin using neuron-specific promoters in the nematode is a clear strength of the genetic model system.

Thanks a lot for these positive remarks.

Weaknesses:(1) The manuscript begins by identifying Calcineurin as a direct substrate of CMK-1 but ends by showing that CMK-1 and Calcineurin mostly act in different neurons to regulate nociceptive habituation which disrupts the logical flow of the manuscript.

We understand this point and we have carefully considered and (reconsidered) the way to articulate the report. However, we could not present the story much differently as we would have no justification to investigate the role of TAX-6 and its interaction with CMK-1, if we would not have first identified it as phospho-target in vitro*.* Carefully considering this point, we found that the abstract of the first manuscript version was probably too cursory and susceptible to trigger wrong expectations among readers. We have thus extensively revised the abstract to clarify this point. Furthermore, we have reinforced this point in the last paragraph of the introduction and in the conclusion paragraph of the Discussion.

(2) The physiological relevance of CMK-1 phosphorylation of Calcineurin is not clear.

We do agree and have explicitly mentioned this aspect in the abstract, in the end of the introduction, and in the discussion section.

(3) It is not clear if Calcineurin is already a known substrate of CaM Kinases in other systems or if this finding is new.

We are not aware of any study having shown Calcineurin is a direct target of CaM kinase I. But it was found to be substrate of CaM kinase II as well as of other kinases, as we explicitly presented in the discussion section. We have complemented the text mentioning we are not aware of Calcineurin having so far been reported to be a CaM kinase I substrate.

**Recommendations for the authors:**

**Reviewer #1 (Recommendations for the authors):**
(1) The authors might consider reorganizing the results, so that the substrate phosphorylation analysis follows the cmk-1 habituation data, as it may not be clear to the reader why you are looking for substrates downstream of cmk-1 at that point. Or the authors could mention the previous habituation data for cmk-1 at the beginning of the results.

Thank you. This is something that we considered while (re-)writing. However, we prefer to keep CMK-1 data side-by-side with TAX-6 data, regarding the result section. Nevertheless, we have modified the last paragraph of intro to better transition and justify the specific interest of searching for CMK-1 targets in the context of the present study.

(2) Line 209: 'controls' is too strong a word. 'regulates' would be better, and it should be stated that this is for 'spontaneous reversal behavior'.

Thank you. This was modified.

(3) Line 359: we suspect that these reflect functional enrichments.

We don’t see what would exactly be wrong with the original sentence. The proposed change (if it is a proposed change) would completely obliterate the intended meaning of our sentence. We rewrote the sentence to be as clear as possible, as follows: ”Even if we cannot rule out an actual inclination of the CaM kinase pathway to regulate these processes, we suspect that these GO term enrichments rather reflect an analytical bias toward abundant proteins.”

(4) Line 563: In this subsection, it is not made clear when the T0 and T60 heat pulses are given, in relation to the 20s ISI heat pulses given for 60 minutes. Are they the first and last pulse, or given some time before or after this train of heat pulses?

Thanks for spotting this poor description, which we have improved in the revised manuscript. The heat pulse recording is given immediately before and immediately after the 60 min of repeated stimulation. After the T0 heat pulse recording there is a period of about 30 s (period of post stimuli recording + transfer from the recording device (INFERNO) to the habituation device (ThermINATOR)). For the T60 acquisition, there is a lag of about 50 s between the last ‘habituation’ stimuli and the recording stimuli (time needed to move the plate between the habituation device and the recording device + 40 s of baseline reversal recording in the absence of heat stimuli).

**Reviewer #2 (Recommendations for the authors):**
(1) There appears to be little to no connection between the phosphorylation site discovered in Calcineurin (S443) and the behavioral phenotypes being studied. What is the thermo-nociceptive response if phosphorylation of S443 in Calcineurin is blocked (using a S443A mutation) and/or combined with CMK-1 gain of function?

Thanks for the suggestion. The suggested analysis is complicated by several factors. First, the *tax-6(lf)* is not directly suitable for rescue analysis (until we would have identified a way to restore baseline reversal), so we cannot use a S443A-carrying rescue transgene. Second, the truncated TAX-6(GF) mutant lacks the C-terminal part, including S443, so we cannot introduce a S443A in this context. The left approach would be to modify the endogenous locus. This again is complicated by the fact that S443 exists in two different isoforms (with conserved RxxS motifs in two different alternative exons). It will be very difficult to perform these experiments until we know more about the expression pattern and function of the respective isoforms. This is work in progress, but this analysis will need to await a future publication.

(2) The authors should state clearly if Calcineurin is a novel substrate of CaM Kinase or if this is already known in the field.

We have complemented the text mentioning we are not aware of Calcineurin having so far been reported to be a CaM kinase I substrate.

(3) The logical flow of the manuscript could be improved given that CMK-1 and Calcineurin appear to act in different cells to regulate nociceptive habituation.

As detailed above, we have considered this point carefully and modified the introduction and the abstract. The discussion about the two places of action was also improved.

(4) More detail about the experimental methods used for the heat-evoked reversals should be included in the Results section.

Thanks for the suggestion. We have improved the description in the Method section and expanded the partial description in the result section, so readers could hopefully proceed without needing to go back and forth with the methods.

(5) Check for typos. For example: line 197 - fix typo "...to a series repeated heat stimulation...".

Thank you. We have carefully read the revised manuscript to correct remaining typos.